# A Self-Explainable Heterogeneous GNN for Relational Deep Learning

**Francesco Ferrini**  *francesco.ferrini@unitn.it*
*University of Trento, Italy*

**Antonio Longa**  *antonio.longa@unitn.it*
*University of Trento, Italy*

**Andrea Passerini**  *andrea.passerini@unitn.it*
*University of Trento, Italy*

**Manfred Jaeger**  *jaeger@cs.aau.dk*
*Aalborg University, Denmark*

**Reviewed on OpenReview:** *https://openreview.net/forum?id=8Q4qxe9a9Z*

## Abstract

Recently, significant attention has been given to the idea of viewing relational databases as heterogeneous graphs, enabling the application of graph neural network (GNN) technology for predictive tasks. However, existing GNN methods struggle with the complexity of the heterogeneous graphs induced by databases with numerous tables and relations. Traditional approaches either consider all possible relational meta-paths, thus failing to scale with the number of relations, or rely on domain experts to identify relevant meta-paths. A recent solution does manage to learn informative meta-paths without expert supervision, but assumes that a node's class depends solely on the existence of a meta-path occurrence. In this work, we present a self-explainable heterogeneous GNN for relational data, that supports models in which class membership depends on aggregate information obtained from multiple occurrences of a meta-path. Experimental results show that in the context of relational databases, our approach effectively identifies informative meta-paths that faithfully capture the model's reasoning mechanisms. It significantly outperforms existing methods in both synthetic and real-world scenarios.

## 1 Introduction

Graph Neural Networks (GNNs) have increasingly become the de-facto standard for many predictive tasks involving networked data, such as physical systems (Sanchez-Gonzalez et al., 2018; Battaglia et al., 2016), Knowledge Graphs (Hamaguchi et al., 2017) and social networks (Wu et al., 2020). By learning effective node embeddings, GNNs offer a unified framework for addressing various graph-based tasks, including node classification, graph classification, and link prediction. However, similar to other representation learning paradigms, GNNs often exhibit a black-box nature in their predictions. Numerous solutions have been proposed for post-hoc explainability of GNN predictions, primarily at the instance-based level (Ying et al., 2019; Vu & Thai, 2020; Miao et al., 2022; Yuan et al., 2021). Yet, as with other deep learning architectures, the ability of these approaches to genuinely reflect the underlying reasoning of the predictor has been called into question (Longa et al., 2024). To tackle this challenge, self-explainable GNNs (Kakkad et al., 2023; Christiansen et al., 2023; Seo et al., 2023) have recently emerged, aiming to ensure that GNN predictions are grounded in interpretable elements, such as subgraphs (Wu et al., 2022; Yu et al., 2020) or prototypes (Zhang et al., 2022; Ragno et al., 2022).

Despite the widespread adoption of GNNs, most approaches are tailored for homogeneous graphs, where edge types are indistinguishable. While encoding edge types as features is a common workaround, the standard solution simply consists in concatenating the one-hot encoded edge type to node features, which eventually boils down to learning an edge-type specific bias. This limitation is particularly problematic in knowledge graphs, which typically feature numerous relations (corresponding to edge types), with only a few being pertinent to a specific predictive task—forming the so-called meta-paths. Existing approaches for heterogeneous GNNs either rely on domain experts to provide relevant meta-paths a priori (Chang et al., 2022; Fu et al., 2020; Li et al., 2021; Wang et al., 2019), or attempt to learn them from data by assigning different weights to various relations (Hu et al., 2020; Lv et al., 2021b; Mitra et al., 2022; Schlichtkrull et al., 2018; Yu et al., 2022; Yun et al., 2019b; 2022a; Zhu et al., 2019), a solution that fails to scale with the number of candidate relations.

Recently, `MP-GNN` (Ferrini et al., 2024) has been proposed as a solution for learning meta-paths without requiring user supervision. This approach employs a scoring function to predict the potential informativeness of partial meta-paths, enabling efficient exploration of the combinatorial space of candidate meta-paths. However, a key limitation of `MP-GNN` is the existential quantification assumption which states that a node's class is primarily dependent on the *existence* of a meta-path instance, meaning just one instance is sufficient for accurate prediction.

its assumption that a node's class is primarily dependent on the *existence* of a meta-path instance, meaning just one instance is sufficient for accurate prediction. While this assumption may be reasonable for knowledge graphs, it is unrealistic when dealing with relational databases, where entities are characterized by numerous categorical and numerical attributes. On the other hand, this same complexity makes relational databases particularly well-suited for GNN technology, as evidenced by the growing interest in what is now being termed relational deep learning (Fey et al., 2023).

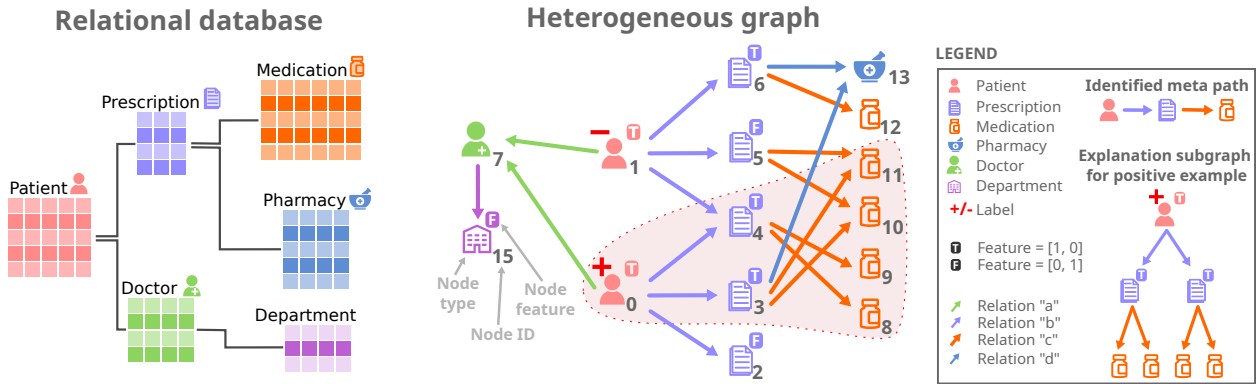

Figure 1: **Left**: Relational database schema for a medical domain. **Right**: Heterogeneous graph representation of (part of) the database. The highlighted subgraph shows a prototypical counts-of-counts pattern characterising positive patients, namely having at least two exempt prescriptions (represented by node feature T), each containing at least two medications. Existing heterogeneous GNNs struggle with these patterns as they need to learn a separate weight matrix for each edge type in the graph, while `MPS-GNN` is capable of learning the relevant meta path without any direct user supervision.

In this paper, we extend the concept behind `MP-GNN` beyond the simple existential quantification of meta-paths. We introduce *Meta-Path Statistics GNN* (`MPS-GNN`), an approach that automatically identifies relevant meta-paths, where the informative content is determined by learnable statistics computed on their realizations. These include counts-of-counts statistics such as having at least two exempt prescriptions with at least two medications each to characterize patients with severe illness, as shown in Figure 1. Note that `MP-GNN` would fail to discriminate between the two patient nodes in the figure, as they both have at least one occurrence of the correct meta-path. An experimental evaluation on both synthetic and real-world relational database tasks demonstrate the significant advantages of the proposed solution over existing alternatives.

Additionally, results show how the meta-path learning strategy behind `MPS-GNN` renders it inherently and genuinely self-explainable, in contrast to many existing self-explainable GNN architectures whose explanations often lack fidelity (Christiansen et al., 2023).

## 2 Related Work

Relational deep learning (Fey et al., 2023) has recently emerged as a paradigm advocating the application of deep learning technology, and GNNs in particular, to relational databases. The rationale behind this research direction is the popularity of relational databases as a mean to store relational information in a variety of application domains, combined with the fact that relational databases can be seen as heterogeneous graphs, with tables converted into sets of nodes and relations into (typed) edges between table entries (Robinson et al., 2024). This transformation allows the application of heterogeneous Graph Neural Networks (GNNs) to this kind of data.

A common characteristics of most heterogeneous graphs, including those deriving from knowledge graphs and relational databases, is that only few relations convey relevant information when targeting a specific predictive task. For this reason, plain GNNs, that do not distinguish between edge types, struggle with these type of graphs. The most popular line of research for heterogeneous GNNs identifies meta-paths, i.e., sequences of relations, as primary sources of information. Existing approaches to incorporate meta-paths follow in two main categories. The former requires domain experts to identify relevant meta-paths for the task at hand (Chang et al., 2022; Fu et al., 2020; Li et al., 2021; Wang et al., 2019). Clearly, this approach is suboptimal as it requires this domain information to be readily available. The alternative solutions, either utilize relation-specific graph convolutions, capturing relational patterns with distinct parameters or dedicated components for each type of relation (Hu et al., 2020; Lv et al., 2021b; Schlichtkrull et al., 2018; Yu et al., 2022) or focuses on graph transformation and multi-view learning to enhance relational representations (Mitra et al., 2022; Yun et al., 2019b; 2022a; Zhu et al., 2019). Although these methods are effective with a limited number of relations, their performance quickly deteriorates as the number of candidate relations grows.

Recently, a novel approach named `MP-GNN` (Ferrini et al., 2024) has been introduced to tackle the aforementioned challenges and automatically learn relevant meta-paths from data. The approach leverages a scoring function predicting the potential informativeness of partial meta-paths to guide the search in the combinatorial space of candidate meta-paths. A major limitation of this approach is the fact that it assumes that the existential quantification of the meta-path is informative for the class label. This assumption makes the approach unsuitable for relational deep learning tasks, in which statistics extracted from table attributes are arguably crucial to characterize predictive targets. Our approach substantially generalizes the `MP-GNN` method, by designing a scoring function that can predict the informativeness of partial meta-paths in terms of the statistics that could be constructed on top of their realizations. This extension is crucial in allowing `MPS-GNN` to be effectively applied to relational deep learning settings, as shown by our experimental evaluation. More details about the comparison between `MP-GNN` and `MPS-GNN` in 4.6.

Explainability in GNN is a major research trend, with plenty of approaches for post-hoc explanation of GNN predictions (instance-based explainability (Ying et al., 2019; Vu & Thai, 2020; Miao et al., 2022; Yuan et al., 2021)) and others aiming to explain the GNN model as a whole (model-based explainability (Chen et al., 2024; Wang & Shen, 2022; Azzolin et al., 2022; Yuan et al., 2020)). Specialized metrics have been developed to estimate the faithfulness of an explanation in relation to the method's input processing behavior (Agarwal et al., 2023; Yuan et al., 2022; Amara et al., 2022; 2023; Zheng et al., 2023). Following a similar evolution in the XAI literature of convolutional neural networks, self-explainable GNNs (Wu et al., 2022; Yu et al., 2020; Zhang et al., 2022; Ragno et al., 2022) have recently been proposed to encourage GNN models to adhere to their explanations by design. However, experimental studies have questioned the faithfulness of the explanations provided by these approaches (Christiansen et al., 2023; Azzolin et al., 2025), highlighting the difficulty of achieving genuine explainability with GNNs. A key advantage of our proposed method is that meta-paths can naturally serve as model-level explanations, making `MPS-GNN` the first truly self-explainable GNN designed for relational deep learning applications. Our experimental evaluation confirms the faithfulness of the explanations to the model's behaviour.

# 3 Preliminaries

This section presents the core concepts that will be utilized in the rest of the paper.

**Definition 3.1** (*Relational Database*). A **relational database** $(\mathcal{T}, \mathcal{L})$ consists of a collection of tables $\mathcal{T} = T_1, ... T_n$ and links between these tables $\mathcal{L} \subseteq T \times T$. Each table is a set $T = \{e_1, ..., e_n\}$ where the elements $e \in T$ are referred to as rows or entities. Each entity is a tuple $e = (\mathcal{P}_e, \mathcal{K}_e, a_e)$ where $\mathcal{P}_e$ is the **Primary Key** that uniquely identifies the entity $e$; $\mathcal{K}_e$ is the set of **Foreign Keys** corresponding to a primary key in other tables, thus connecting the tables; $a_e$ corresponds to the **Attributes** of the entity $e$.

**Definition 3.2** (*Heterogeneous graph*). A **heterogeneous graph** is a directed graph $\mathcal{G} = (\mathcal{V}, \mathcal{E}, X_\mathcal{V})$ where $\mathcal{V}$ is the set of nodes or entities, $\mathcal{E}$ is the set of directed edges (graphs induced by relational databases will be inherently directed) and $X_\mathcal{V}$ is a matrix of node attributes (with $x_v$ being the attribute vector of node $v$). Each edge is represented as a triple $(u, r, v)$, indicating that nodes $u$ and $v$ are connected via relation $r$ (written as $u \xrightarrow{r} v$). We indicate the set of relations in the graph as $\mathcal{R}$.

For a node $v$ and a relation $r$ we denote with $\mathcal{N}_v^r$ the set of nodes that can be reached from $v$ by following relation $r$. We refer to this set as $r$-neighbors.

**Definition 3.3** (*Meta-path*). A meta-path $mp$ is a sequence of relations defined on a heterogeneous graph $\mathcal{G}$, represented as $\xrightarrow{r_1} \xrightarrow{r_2} \ldots \xrightarrow{r_L}$, where $r_1, \ldots, r_L$ are relations in $\mathcal{R}$. knowledge graphs.

**From relational database to heterogeneous graph** A relational database can be interpreted as an heterogeneous graph where row $e$ becomes node $v$; attributes $a_e$ become node features $x_v$; links $\mathcal{L}$ between entries of two tables are identified by the pair of primary $\mathcal{P}$ and foreign $\mathcal{K}$ keys in the two tables. Each pair of connected tables, originates a relation in the graph, specified by $r$.

# 4 Methodology

For now we restrict attention to binary node classification problems. Our main problem is to construct meta-paths $\xrightarrow{r_1} \xrightarrow{r_2} \ldots \xrightarrow{r_L}$ that are predictive features for the class label. When considering a meta-path as a feature, we are thinking of possible numerical features that can be defined by collecting and aggregating information that is found along all occurrences of the meta-path in a concrete data graph, such as the count-of-count feature illustrated in Figure 1. We construct meta-paths following a strategy that is

- *Greedy*: a partially constructed meta-path $\xrightarrow{r_1} \xrightarrow{r_2} \ldots \xrightarrow{r_i}$ is extended by a next relation $\xrightarrow{r_{i+1}}$ without lookahead for possible completions $\xrightarrow{r_{i+2}} \ldots \xrightarrow{r_L}$. Similar as in Ferrini et al. (2024), we try to estimate the *potential informativeness* of nodes reached by $\xrightarrow{r_{i+1}}$ by learned weights associated with the nodes.

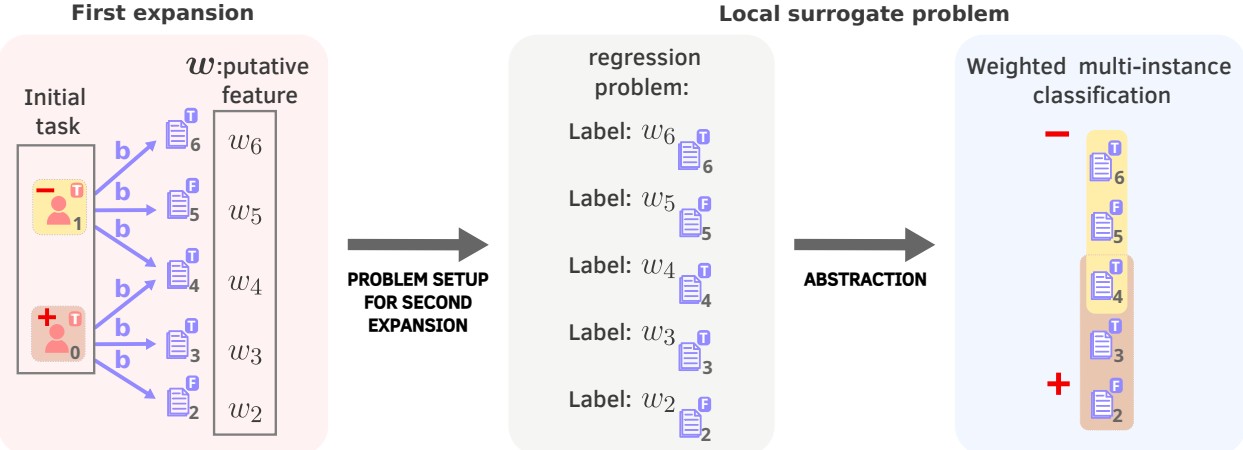

Figure 2: Outline of greedy local meta-path construction

These weights represent putative features that can either be directly materialized as functions of the nodes' attributes, or that can be constructed as features of meta-path extensions starting at the node.

- *Local*: the meta-path construction step $\xrightarrow{r_i}\xrightarrow{r_{i+1}}$ is performed based only on local consideration of the nodes reached by $r_i$, and their $r_{i+1}$ successors. The already constructed meta-path prefix $\xrightarrow{r_1}\xrightarrow{r_2}\ldots\xrightarrow{r_i}$ plays no explicit role in this step. We realize this locality by defining for each step a surrogate classification task for the nodes reached by $r_i$. The problem of extending the constructed meta-path prefix then translates into the problem of finding the first relation for a relevant meta-path solving the surrogate problem. The surrogate problems take the form of *weighted multi-instance classification* tasks.

Figure2 illustrates the high-level principles of our approach. Given an initial (binary) node classification task, a first relation is identified that could solve the task with the help of a putative node feature (weight) $\boldsymbol{w}$ on the successor nodes. Then a surrogate classification task is set up whose target is to materialize the putative feature as a feature computable from the data. This surrogate task takes the form of a weighted multi-instance classification task, which can be seen as an abstraction of a direct regression problem with target $\boldsymbol{w}$ (see Section 4.2).

Note that both the greedy and local properties mirror core principles of growing decision trees, which are built incrementally, adding one relation at a time based on solving local classification sub-tasks. The improvements in time complexity with respect to an algorithm that looks at every possible relations in the meta-path construction are detailed in section 4.2.1. In Section 4.1, we describe the *weighted multi-instance classification* task and its application in our scenario. Section 4.2 outlines the methodology for constructing meta-paths by scoring graph relations, including the examples based on Figure 1. In Section 4.3, we detail the GNN used in our setup, and Section 4.4 presents the complete framework of the proposed approach.

## 4.1 Weighted multi-instance classification

We consider a variant of multi-instance classification, where each instance consists of a bag $\mathcal{B}$ of nodes with a class label in $\{+,-\}$, and each node $v \in \mathcal{B}$ is assigned a weight $\alpha(v,\mathcal{B})$. We denote with $\mathcal{S}^+, \mathcal{S}^-$ (training) sets of positively and negatively labeled bags, respectively. The intention is to interpret the label of the bag as a function of element-level features, and that $\alpha(v,\mathcal{B}) \in \mathbb{R}$ represents a weight of the contribution of $v$'s feature value to the class label of $\mathcal{B}$. This weight can be positive or negative, and its absolute value can be interpreted as a measure for the importance of node $v$ in bag $\mathcal{B}$ (which may differ for different bags that $v$ is an element of). See Foulds & Frank (2010) for related generalizations of the standard multi-instance learning setting.

Our goal is to predict the bag label via discriminant functions of the form

$$F(\mathcal{B}) = \sum_{v \in \mathcal{B}} \alpha(v,\mathcal{B}) f(v), \tag{1}$$

where $f(v)$ is a learnable node feature function. Specifically, we consider functions that are parameterized by a relation $r$, and are of the form

$$f(v,r,\Theta,\boldsymbol{w}) = \begin{cases} \Theta^T x_v & \text{if } \mathcal{N}_v^r = \emptyset \\ \Theta^T x_v \sum_{u \in \mathcal{N}_v^r} w_u & \text{if } \mathcal{N}_v^r \neq \emptyset \end{cases} \tag{2}$$

where $x_v$ denotes the attribute vector of $v$, $\Theta$ is a trainable parameter vector, and $\boldsymbol{w}$ is a vector of trainable weights $w_u \in [0,1]$ assigned to $v$'s $r$-neighbors. The node feature function is thus computed as a combination of $v$'s own attributes and the putative feature $w_u$ of its successors. When the node attributes $x_v$ are not informative for solving the multi-instance classification task, then parameters $\Theta$ can be learned that make $\Theta^T x_v$ constant for all $v$, and thus this part of the node feature function becomes irrelevant (a suitable $\Theta$ exists e.g. under the mild assumption that $x_v$ contains at least one categorical attribute [1] in a one-hot

---

[1]In a heterogeneous graph built from a relational database, this would be the table the node belongs to. If node representations are simply embeddings, one can achieve the same goal by concatenating a constant feature to the embedding.

encoding: then $\Theta^T x_v = 1$ for $\Theta$ that is set to 1 for all entries corresponding to the one-hot encoding of the categorical attribute, and zero everywhere else). The factor $\sum_{u \in \mathcal{N}_v^r} w_u$ captures a dependence of $f(v)$ on the $r$-neighborhood of $v$. When the $r$-neighborhood is non-informative because neither the number nor the identity of nodes $u \in \mathcal{N}_v^r$ has discriminative value, then the $\sum_{u \in \mathcal{N}_v^r} w_u$ factor can be made irrelevant by learning constant weights $w_u$.

Ideally, the discriminant function separates the classes in the sense that for any pair $\mathcal{B}^+ \in \mathcal{S}^+, \mathcal{B}^- \in \mathcal{S}^-$ we have $F(\mathcal{B}^+) > F(\mathcal{B}^-)$. We note that this problem would be trivially solvable e.g. in the case where every positive bag contains a node $v$ that has an $r$-successor, which is not also an $r$-successor of some node in a negative bag. Then assigning a weight of 1 to all such $r$-successor nodes, and a weight of 0 to all other nodes, would separate the classes. In reality, the complex connectivity of relations will preclude such simple solutions, and perfect separation in general. We therefore use as our learning objective the relaxed loss function

$$L(r, \Theta, \boldsymbol{w}) = \sum_{\mathcal{B}^+ \in \mathcal{S}^+, \mathcal{B}^- \in \mathcal{S}^-} \sigma\Big(F(\mathcal{B}^-, r, \Theta, \boldsymbol{w}) - F(\mathcal{B}^+, r, \Theta, \boldsymbol{w})\Big), \tag{3}$$

where $\sigma$ is the sigmoid function. In practice, given that the number of terms in the sum is quadratic in the number of training examples, we approximate (3) by a random sample of positive and negative bags.

## 4.2 Relation Scoring

The initial weighted multi-instance classification problem for our meta-path construction process is defined by letting each positive (negative) target node $v$ form a one-element bag $\{v\}$ with the corresponding label, and weight $\alpha(v, \{v\}) = 1$. Denote with $\mathcal{S}_1^+, \mathcal{S}_1^-$ the sets of all initial positive and negative bags, respectively. At all iterations we select the relation that minimizes the loss (also referred to as the scoring function)

$$L(r) = \min_{\Theta, \boldsymbol{w}} L(r, \Theta, \boldsymbol{w}). \tag{4}$$

If all candidate relations fail to minimize the loss, i.e., optimizing $\Theta, \boldsymbol{w}$ does not lead to substantially lower loss than using random parameters (i.e., does not improve by at least 30%), then the meta-path construction terminates and the current meta-path is returned. Otherwise, the current meta-path is extended with the minimal loss relation $r$. At this point, the problem becomes capturing the putative node features $\boldsymbol{w}$ by actual features.

A possible approach would be to set this up as a node regression task with target values $w_u$. However, due to the often very large set of alternative optimal solutions $\boldsymbol{w}$ in the minimization (4), this would lead to a too restrictive task. Our goal is to approximate the whole space of possible regression tasks defined by alternative $\boldsymbol{w}$ as a single weighted multi-instance classification task. For this, let $r_i$ denote the relation found to minimize (4) in iteration $i$ with optimal parameters $\Theta_i$. For each positive bag $\mathcal{B}^+ \in \mathcal{S}_i^+$ define the new bag

$$\mathcal{B}_{new}^+ = \cup_{v \in \mathcal{B}^+} \mathcal{N}_v^{r_i} \tag{5}$$

containing the $r_i$-children of the nodes in $\mathcal{B}^+$. Similarly for negative bags. For $u \in \mathcal{B}_{new}^+$ define the node weight by the following sum over all nodes $v \in \mathcal{B}^+$ that have $u$ as an $r_i$-neighbor:

$$\alpha(u, \mathcal{B}_{new}^+) = \sum_{\{v \in \mathcal{B}^+ : u \in \mathcal{N}_v^{r_i}\}} \Theta_i^T x_v \alpha(v, \mathcal{B}^+). \tag{6}$$

This definition of the weights enables to essentially ignore in the setup of the multi-instance classification task for iteration $i+1$ those nodes $u$ that are $r_i$-successors only of nodes $v$ that did not play a major role in solving the task of the previous iteration – either because of a low absolute value of $\Theta_i^T x_v$, or because $\alpha(v, \mathcal{B}_i^+)$ already had a low absolute value. The sets of all $\mathcal{B}_{new}^+ (\mathcal{B}_{new}^-)$ form the new training sets $\mathcal{S}_{i+1}^+ (\mathcal{S}_{i+1}^-)$.

**Toy Example:** To provide a clear understanding of the proposed approach, we illustrate it with the toy example of Figure 1, where patients are positive if, and only if, they have at least two exempt prescriptions, each containing at least two medications.

Initially, the approach evaluates the *potential informativeness* of two relations originating from patients: relation $a$ and relation $b$. Figure 3a illustrates the scoring process for relation **a**. The first step involves instantiating the node feature functions for patient nodes 0 and 1 by applying Equation 2. Since both nodes share the same unique neighbor, the loss function 4 reduces to the constant $\frac{1}{2}$, which cannot be minimized by $\Theta$ and $\boldsymbol{w}$.

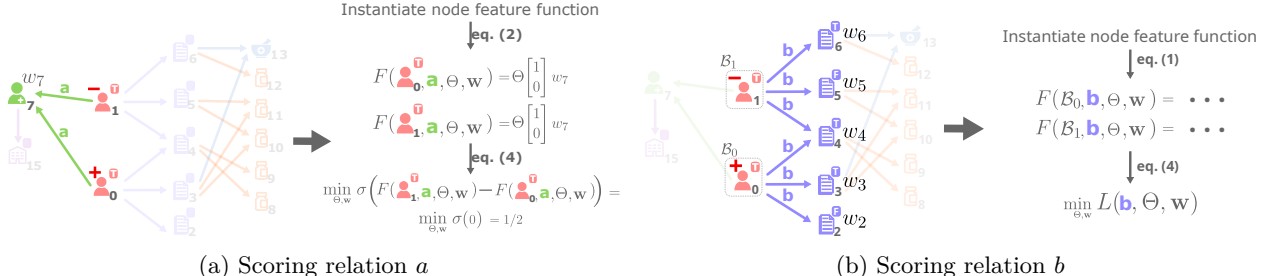

(a) Scoring relation $a$            (b) Scoring relation $b$

Figure 3: Scoring the first two relations

Similarly, relation $b$ is scored. The loss function 4 now becomes

$$\sigma(\Theta^T x_1(w_4 + w_5 + w_6) - \Theta^T x_0(w_2 + w_3 + w_4)),$$

where $x_0 = x_1 = (1,0)^T$ is the attribute vector representing the T value in a one-hot encoding. This loss can be brought arbitrarily close to zero, e.g. by $w_6 \to 0$, $w_5 \to 0$, $w_3 \to 1$, $w_2 \to 1$, and $\Theta = (Z, 0)^T$ with $Z \gg 0$. Consequently, relation $b$ scores higher than relation $a$ and is selected for further extension. Note that the low loss of relation $b$ is entirely due to its potential informativeness: the $b$ neighborhoods of the nodes 0 and 1 are completely isomorphic, and therefore the meta path consisting of $b$ alone provides no discriminative information.

To extend the meta path prefix $b$ we set up the weighted multi-instance classification task for the second iteration. This gives us a positive bag $\mathcal{B}_2$ containing nodes $\{2, 3, 4\}$ and a negative bag $\mathcal{B}_3$ containing nodes $\{4, 5, 6\}$. Node weights for the bags are computed according to Equation (6), and here simply yield uniform weights $\alpha(k, \mathcal{B}_2) = Z$ for all $k \in \{2, 3, 4\}$, and $\alpha(k, \mathcal{B}_3) = Z$ for all $k \in \{4, 5, 6\}$. Note that node 4 has separate weights for the two bags it is part of.

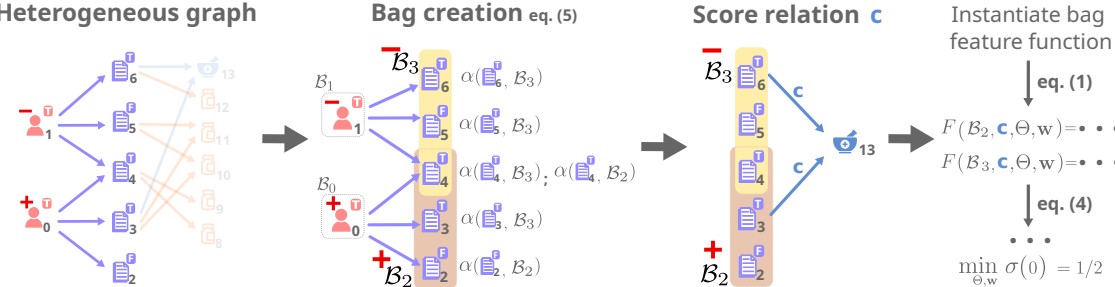

Figure 4: Bag generation and scoring of relation $c$.

For solving the new classification task we score the candidate relations $c$ and $d$. The right part of Figure 4 illustrates the scoring of $c$. Due to the indistinguishable structure of the $c$-neighborhood for bags $\mathcal{B}_2$ and $\mathcal{B}_3$ the loss function 4 now reduces again to the constant $1/2$, as in the scoring of relation $a$.

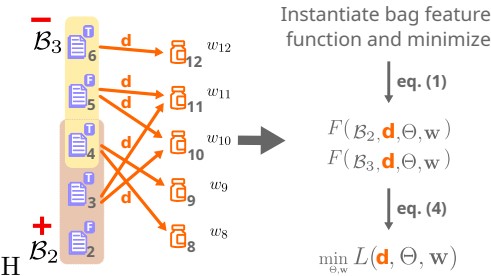

Figure 5: Scoring relation $d$.

For scoring the relation $d$ we obtain as the loss function 4

$$\sigma\left(Z\Theta^T\begin{bmatrix}1\\0\end{bmatrix}(w_{12}-w_{10}-w_{11})+Z\Theta^T\begin{bmatrix}0\\1\end{bmatrix}(w_{10}+w_{11}-1)\right)$$

(see details in Appendix A.1). This loss can be brought arbitrarily close to zero, e.g. by $w_{10}, w_{11} \to 1$, $w_{12} \to 0$ and $\Theta = (Z',0)^T$ with $Z' \gg 0$. Thus, relation $d$ is selected to extend the meta-path. The constructed meta-path $\xrightarrow{b}\xrightarrow{d}$ now is sufficient to discriminate between positive and negative examples. This is not directly visible at this step in the meta-path construction process, but will be found by training an `MPS-GNN` on this meta-path, as detailed in sections 4.3 and 4.4. Check A.10 for a similar example with more complex node features

### 4.2.1 Time complexity analysis

The purpose of the scoring function is to iteratively construct the meta-path that best classifies the target nodes, avoiding the need to explore all possible paths. A naive search algorithm that simply tests all possible paths would have a polynomial complexity of $O(|\mathcal{R}|^L)$, where $|\mathcal{R}|$ is the number of relations in the graph, and $L$ is the maximum length of the meta-path we aim to find. By leveraging the scoring function, the complexity is reduced to linear $O(|\mathcal{R}| \cdot L)$. This improvement is achieved because, at each step, one relationship is added to the meta-path under construction, and the subsequent search builds upon it without reconsidering the other relationships scored in the same iteration.

### 4.3 `MPS-GNN`

Similarly to `MP-GNN`, in the `MPS-GNN` framework a meta-path $r_1, ..., r_L$ defines a multi-relational GNN with $L$ layers. In this setup, each layer of the network corresponds to a specific relation in the meta-path: the initial layer is linked to the final relation $r_L$, progressing sequentially until the last layer, which corresponds to $r_1$. Our forward model then takes the form:

$$h_v^{(l+1)} = \sigma\left(W_0^{(l)}h_v^{(l)} + W_{neigh}^{(l)}\sum_{u\in\mathcal{N}_v^{r_{L-l}}}h_u^{(l)} + W_1^{(l)}h_v^{(0)}\right) \tag{7}$$

where $\mathcal{N}_v^{r_{L-l}}$ are neighbours of node $v$ under relation $r_{L-l}$, $h_v^{(l)}$ is the embedding of node $v$ in layer $l$, $h_v^{(0)} = x_v$ is the feature vector of node $v$, while $W_0^{(l)}$, $W_{neigh}^{(l)}$ and $W_1^{(l)}$ are learnable parameter vectors. Note that the latter term $W_1^{(l)}h_v^{(0)}$, which is missing the original `MP-GNN` (Ferrini et al., 2024), represents a skip connection between the input and the $l+1$ layer. This allows the network to access the node attributes at each layer, which is essential for enabling the `MPS-GNN` to capture node features corresponding to the $\Theta \cdot x_v$ terms in the meta-path construction as shown in the ablation study in A.2). As in Ferrini et al. (2024), the definition can be generalized to multiple meta-paths by concatenating the embeddings obtained from each of them using $h_v^{(l+1)} = \big\|_{k=1}^K h_{(v,k)}^{(l+1)}$ where $K$ is the number of meta-paths, $h_{(v,k)}^{(l+1)}$ is the embedding of node $v$ according to meta-path $k$ and $\|$ is the concatenation operator.

## 4.4 The overall algorithm

Algorithm 1 outlines the whole `MPS-GNN` procedure for the single meta-path case (in practice, a $K$ beam search is used and multiple meta-paths are learned). The algorithm takes as input a graph $\mathcal{G}$, the set of available relations $\mathcal{R}$, an initial set of binary node labels $\mathcal{Y}$ a maximal meta-path length $L_{MAX}$ and a stopping criteria value $\eta$. It initializes the targets $\mathcal{S}$ with a set of singletons (one per labelled node) and their alpha values (collectively indicated by $\mathcal{A}$) to 1. At each iteration, the scoring function identifies the relation minimizing Eq. 4 and appends it to the meta-path $mp$. If no relation improves the loss by a factor $\eta$, the algorithm stops and returns the meta-path $mp^*$ constructed so far. The algorithm then evaluates $mp$ by training `MPS-GNN` on node labels, and tests it using the $F_1$ score (computed on a validation set, omitted for brevity), which reflects the meta-path's performance *when embedded* in an `MPS-GNN`, as opposed to its potential informativeness measured by the scoring function. For training `MPS-GNN`, node embeddings are updated using 7.

---

**Algorithm 1** MPS-GNN LEARNING

**procedure** LEARNMPS-GNN($\mathcal{G}, \mathcal{R}, \mathcal{Y}, L_{MAX}, \eta$)
  $\quad mp^* \leftarrow [\,], F_1^* \leftarrow 0, \mathcal{S} \leftarrow \mathcal{Y}, \mathcal{A} \leftarrow 1$
  $\quad$**while** $|mp| < L_{MAX}$ **do**
  $\quad\quad r^* \leftarrow argmin_r L(r)$ $\qquad\qquad$ ▷ Eq. 4
  $\quad\quad$**if** $\min_{r \in \mathcal{R}} L(r) \geq \eta L_{\text{init}}(r)$ **then**
  $\quad\quad\quad$**return** $mp^*$
  $\quad\quad$**end if**
  $\quad\quad mp \leftarrow mp, r^*$
  $\quad\quad gnn \leftarrow$ TRAIN(MPS-GNN($mp$), $\mathcal{G}, \mathcal{Y}$)
  $\quad\quad F_1 \leftarrow$ TEST($gnn$)
  $\quad\quad$**if** $F_1 > F_1^*$ **then**
  $\quad\quad\quad mp^* \leftarrow mp, \ F_1^* \leftarrow F_1$
  $\quad\quad$**end if**
  $\quad\quad \mathcal{A}, \mathcal{S} \leftarrow$ NEW-TARGETS($\mathcal{S}, r^*$) $\quad$ ▷ Eq. 5, 6
  $\quad$**end while**
  $\quad$**return** $mp^*$
**end procedure**

---

The algorithm keeps track of the best meta-path prefix together to its $F_1$ score, and creates new target bags and $\alpha$ values as specified in Equations 5 and 6 for the next relation scoring round. The algorithm ends when the maximum meta-path length $L_{MAX}$ is reached The algorithm is described for simplicity in the context of a single meta-path and without the additional stopping criteria described in section 4.2. In practice, however, the implementation employs a beam search over the meta-path space, selecting the top $K$ best-scoring meta-paths concatenating their embedding for the final node representation as detailed at the end of section 4.3. MPS-GNN scales *linearly* in the number of relations and nodes, thanks to its incremental construction of meta-paths. The value chosen for $L_{\text{MAX}}$ in the experiment was 4, while $\eta$ was set to 0.7. The method is however quite robust to variations in $\eta$, as the identified meta-paths remained unchanged for a wide range of values in our experiments.

## 4.5 `MPS-GNN` is a self-explainable model

Self-explainable GNNs (Kakkad et al., 2023; Christiansen et al., 2023) are a class of GNN models that aims to achieve explainability by-design. At a high level, these models can be seen as composed of two modules: a *detector* that extracts a class-discriminative subgraph, and a *classifier* that outputs a prediction based on the extracted subgraph. By relying on meta-paths for its predictions, `MPS-GNN` is a self-explainable GNN model. The scoring function serves as the detector, identifying relevant meta-paths, while the network built using them acts as the classifier. By construction, the network can only access the meta-path induced subgraph, making it strictly sufficient by construction (no changes outside the meta-path induced graph affect the prediction). An analysis of the faithfulness of `MPS-GNN`'s explanations is provided in Section 5.3. While most approaches for GNN explainability focus on the topological aspect, by identifying (hard or soft) subgraphs as explanations, determining which node features contribute to the prediction is also relevant from an interpretability perspective. While interpretability at the node feature level could be encouraged by introducing sparsifying norms for the learnable parameter vectors in Eq. 7, guaranteeing a fully transparent processing of node features in the layerwise node embedding computation is beyond the scope of current GNN-based architectures.

## 4.6 Comparison with `MP-GNN`

The novelty of our approach compared to `MP-GNN` lies in our model's ability to learn meta-paths that are relevant to the target node class, not merely based on their existence but on statistical measures related

to their occurrences. In practice the key difference between `MPS-GNN` and `MP-GNN` lies in the way in which relations are scored. Specifically `MP-GNN` predicts the class label $y$ of a node $v$, in the first iteration, with $\tilde{y}_v^r = \Theta^T x_v \max_{u \in \mathcal{N}_v^r} w_u$ where the *max* aggregation of the neighbours is used, indicating that a candidate relation $r$ is informative for the label of a node $v$ if at least one of the neighbors $\mathcal{N}_v^r$ of $v$ according to $r$ belongs to the ground-truth meta-path, and $v$ has the right features. To leverage meta-path occurrences, `MPS-GNN` employs a *sum* aggregation strategy, as detailed in Eq. 2, enabling the counting of their occurrences.

In subsequent iterations, `MP-GNN` also employs a bag creation process, where positive bags are formed by including neighbors of positive nodes that are predicted to be positive at least once across multiple prediction procedures. This formulation ensures that the node responsible for the positive label, and thus aligned with the correct meta-path, remains in a positive bag. Unlike this approach, however, we must ensure that all neighbor nodes involved in multiple occurrences of the *correct* meta-path are included in a positive bag. As detailed in eq. 5, the neighbors of positive nodes are therefore placed in a positive bag without the need of any additional prediction step.

The enhancements introduced in `MPS-GNN`, compared to `MP-GNN`, enable it to handle predictions over relational databases where the class label may depend on statistical measures derived from meta-path occurrences.

## 5 Experiments

Our experimental evaluation seeks to address the following research questions:

**Q1** Can `MPS-GNN` recover the correct meta-path when increasing the setting complexity?

**Q2** Does `MPS-GNN` outperform existing approaches in tasks over real world relational databases?

**Q3** Is `MPS-GNN` self-explainable?

We compared `MPS-GNN` with approaches that don't require predefined meta-paths, handle numerous relations, and incorporate node features in learning. The identified competitors include: `MLP`, to test the sufficiency of target node features alone; `GCN` (Kipf & Welling, 2016), a baseline non-relational model; `RGCN` (Schlichtkrull et al., 2017), extending `GCN` for multi-relational graphs, with distinct parameters for each edge type; `HGN` (Lv et al., 2021a), a heterogeneous GNN model extending `GAT` for multiple relations; `GTN` (Yun et al., 2019a), which transforms input graphs into different meta-path graphs where node representations are learned; `Fast-GTN` (Yun et al., 2022b), an optimized `GTN` variant; `R-HGNN` (Yu et al., 2021), a relation-aware GNN using cross-relation message passing; and `MP-GNN` (Ferrini et al., 2024), the original meta-path GNN supporting only existentially quantified meta-paths.

We implemented our model using PyTorch Geometric, and used the competitors' code from their respective papers for comparison. For training `MPS-GNN`, we used a 70/20/10 split for training, validation, and testing, respectively, and reported the test results for the model selected based on its validation performance. For the sake of comparison with Ferrini et al. (2024), we set the maximum meta-path length to 4 and the beam size to 3. We employed $F_1$ as evaluation metric to account for the unbalancing in many of the datasets. The code is freely available at [2]. Hyperparameters of competitors and `MPS-GNN` can be found in Table 8 in the Appendix.

### 5.1 Q1: `MPS-GNN` consistently identifies the correct meta-path in count based synthetic scenarios

In order to address the first research question, we designed a sequence of synthetic node classification scenarios where the correct structure to be learnt is known. In each scenario, a node is labelled as positive if it is the starting point of at least $c$ occurrences of a given meta-path of length $l$, and negative otherwise. Crucially, existential quantification of meta-paths (as modelled by `MP-GNN` (Ferrini et al., 2024)) is insufficient here, as nodes which are starting points of less than $c$ meta-path occurrences are labelled as negatives. We designed scenarios of increasing complexity by changing the length of the ground-truth meta-path $l$, the number of

---

[2]https://github.com/francescoferrini/MPS-GNN

occurrences $c$, and the overall number of relations $r$ in the dataset. See Figure 6 for the statistics of the different scenarios (left), and for a prototypical example for $l = 2$ and $c = 3$ (right).

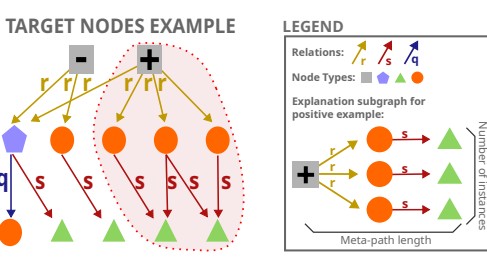

| | **S1** | **S2** | **S3** | **S4** | **S5** | **S6** | **S7** | **S8** |
|---|---|---|---|---|---|---|---|---|
| $\|R\|$ | 5 | 5 | 5 | 5 | 10 | 10 | 10 | 10 |
| $c$ | 2 | 2 | 3 | 4 | 2 | 2 | 3 | 4 |
| $l$ | 2 | 3 | 3 | 2 | 2 | 3 | 3 | 2 |

Figure 6: **(Left)** sample scenario. Nodes are labeled as positive if and only if they are the starting point of at least $c = 3$ instances of the $l = 2$ meta-path "*grey node* $\overset{r}{\to}$ *orange node* $\overset{s}{\to}$ *green node*". **(Right)** statistics of synthetic datasets, with $|\mathcal{R}|$ total number of relations, $c$ number of (correct) meta-path instances in positive nodes, $l$ meta-path length. On top, the explanation subgraph for each dataset. The complexity given by $|\mathcal{R}|$ that is visible from the table.

Table 1 shows the $F_1$ score of each model for an increasing complexity of the classification scenario. Results clearly show that this experimental setting is challenging for existing solutions. While the poor performance of `MLP`, which completely ignores the topological structure, and `GCN`, which ignores the difference between relations, are expected, solutions specifically conceived for heterogeneous networks also struggle with these datasets. Models like `R-HGNN`, `HGN`, `GTN`, and `Fast-GTN`, despite accounting for different relations in the graph, are affected by both the imbalance between positive and negative target nodes and the limited number of instances of neighbors of a certain type. `RGCN` and `MP-GNN` achieve better performance but are still suboptimal. The former, like other relational methods, takes into account the diversity of relations in the graph but still uses all of them, thus struggling to single out the relevant portion of the graph. `MP-GNN`, on the other hand, suffers from its existential quantification assumption, and fails to find the correct meta-path in all scenarios. Conversely, `MPS-GNN` manages to achieve nearly-optimal performance in all scenarios, substantially outperforming all existing strategies[3]. Note that the lookahead capabilities of the scoring function are crucial to the effectiveness of `MPS-GNN`. Appendix A.7 shows how replacing the scoring function with a simple greedy approach leads to failure in learning the correct meta-path. These results allow us to answer the first research question in the affirmative.

Table 1: $F_1$ metric with standard deviations for synthetic datasets

| | **S1** | **S2** | **S3** | **S4** | **S5** | **S6** | **S7** | **S8** |
|---|---|---|---|---|---|---|---|---|
| `MLP` | $0.46_{(\pm0.00)}$ | $0.44_{(\pm0.00)}$ | $0.48_{(\pm0.00)}$ | $0.47_{(\pm0.00)}$ | $0.44_{(\pm0.00)}$ | $0.51_{(\pm0.00)}$ | $0.45_{(\pm0.00)}$ | $0.47_{(\pm0.00)}$ |
| `GCN` | $0.46_{(\pm0.00)}$ | $0.46_{(\pm0.02)}$ | $0.48_{(\pm0.03)}$ | $0.52_{(\pm0.05)}$ | $0.44_{(\pm0.00)}$ | $0.48_{(\pm0.00)}$ | $0.46_{(\pm0.00)}$ | $0.48_{(\pm0.00)}$ |
| `RGCN` | $0.78_{(\pm0.02)}$ | $0.87_{(\pm0.03)}$ | $0.86_{(\pm0.03)}$ | $0.81_{(\pm0.02)}$ | $0.86_{(\pm0.03)}$ | $0.77_{(\pm0.01)}$ | $0.91_{(\pm0.00)}$ | $0.79_{(\pm0.01)}$ |
| `R-HGNN` | $0.50_{(\pm0.00)}$ | $0.44_{(\pm0.03)}$ | $0.47_{(\pm0.01)}$ | $0.47_{(\pm0.04)}$ | $0.53_{(\pm0.00)}$ | $0.48_{(\pm0.01)}$ | $0.46_{(\pm0.02)}$ | $0.48_{(\pm0.02)}$ |
| `HGN` | $0.45_{(\pm0.00)}$ | $0.46_{(\pm0.00)}$ | $0.50_{(\pm0.03)}$ | $0.46_{(\pm0.00)}$ | $0.46_{(\pm0.00)}$ | $0.48_{(\pm0.00)}$ | $0.45_{(\pm0.03)}$ | $0.40_{(\pm0.12)}$ |
| `GTN` | $0.46_{(\pm0.00)}$ | $0.52_{(\pm0.00)}$ | $0.49_{(\pm0.00)}$ | $0.48_{(\pm0.00)}$ | $0.44_{(\pm0.00)}$ | $0.47_{(\pm0.00)}$ | $0.49_{(\pm0.00)}$ | $0.47_{(\pm0.00)}$ |
| `Fast-GTN` | $0.46_{(\pm0.00)}$ | $0.48_{(\pm0.00)}$ | $0.51_{(\pm0.00)}$ | $0.49_{(\pm0.00)}$ | $0.44_{(\pm0.00)}$ | $0.46_{(\pm0.00)}$ | $0.53_{(\pm0.00)}$ | $0.47_{(\pm0.00)}$ |
| `MPGNN` | $0.84_{(\pm0.09)}$ | $0.82_{(\pm0.13)}$ | $0.85_{(\pm0.10)}$ | $0.95_{(\pm0.02)}$ | $0.89_{(\pm0.06)}$ | $0.79_{(\pm0.03)}$ | $0.84_{(\pm0.06)}$ | $0.71_{(\pm0.21)}$ |
| `MPS-GNN` | $\mathbf{0.98}_{(\pm0.00)}$ | $\mathbf{0.98}_{(\pm0.01)}$ | $\mathbf{0.99}_{(\pm0.10)}$ | $\mathbf{0.98}_{(\pm0.00)}$ | $\mathbf{0.99}_{(\pm0.00)}$ | $\mathbf{0.93}_{(\pm0.10)}$ | $\mathbf{0.94}_{(\pm0.00)}$ | $\mathbf{0.95}_{(\pm0.00)}$ |

### 5.2 Q2: `MPS-GNN` surpasses competitors in real world databases, learning relevant meta-paths

Our approach is particularly useful for predictive tasks in relational databases with multiple tables, where features for a target entity may involve statistics from related tables. To address the second research question,

---

[3]The residual error for `MPS-GNN` is due to the fact that despite relying on the correct meta-path, it occasionally leverages spurious instances where the relation sequence is correct but (some of) the node features are not.

we thus focused on three relational databases with many tables: **EICU**, a medical database with 31 tables, where we predict patient stay duration in the eICU, modeled as binary node classification by thresholding duration at 20 hours to achieve two balanced classes.; **MONDIAL**, a geographic database where the task is predicting whether a country's religion is Christian; and **ErgastF1**, containing Formula 1 data, where the task is predicting the winner of a race in a binary classification task where target nodes are represented by a combination of race and pilot. The databases were transformed into graphs as explained in Section 3; for disconnected components, we enhanced connectivity by clustering rows of auxiliary tables. Additional details for the datasets and the procedure are in the Appendix A.3.

Table 2: $F_1$-score with standard deviations of our method and competitors on real-world datasets.

|          | EICU | MONDIAL | ErgastF1 |
|----------|------|---------|----------|
| MLP      | $0.53_{(\pm0.02)}$ | $0.52_{(\pm0.00)}$ | $0.50_{(\pm0.00)}$ |
| GCN      | $0.89_{(\pm0.00)}$ | $0.60_{(\pm0.02)}$ | $0.50_{(\pm0.01)}$ |
| RGCN     | $0.70_{(\pm0.00)}$ | $0.53_{(\pm0.08)}$ | $0.57_{(\pm0.01)}$ |
| R-HGNN   | $0.61_{(\pm0.00)}$ | $0.61_{(\pm0.01)}$ | $0.72_{(\pm0.02)}$ |
| HGN      | $0.75_{(\pm0.00)}$ | $0.72_{(\pm0.01)}$ | $0.70_{(\pm0.04)}$ |
| GTN      | $0.56_{(\pm0.02)}$ | $0.38_{(\pm0.01)}$ | $0.60_{(\pm0.01)}$ |
| Fast-GTN | $0.46_{(\pm0.03)}$ | $0.39_{(\pm0.04)}$ | $0.60_{(\pm0.03)}$ |
| MP-GNN   | $0.87_{(\pm0.02)}$ | $0.36_{(\pm0.06)}$ | $0.71_{(\pm0.01)}$ |
| MPS-GNN  | $\mathbf{0.92}_{(\pm0.01)}$ | $\mathbf{0.74}_{(\pm0.01)}$ | $\mathbf{0.83}_{(\pm0.02)}$ |

**Results** Table 2 presents the $F_1$ scores of `MPS-GNN` and its competitors across three real-world databases, averaged over 5 runs with different seeds. The poor performance of `MLP` clearly indicates that using target node features only is insufficient for classification. Plain `GCN`, which treats the graph as homogeneous, performs well only on the **EICU** dataset, where node degree differences exist between positive and negative nodes. Heterogeneous GNN methods also struggle with these datasets, especially **MONDIAL**, where most approaches fail to outperform a simple `MLP`, and only one (`HGN`) manages to substantially outperform the non-heterogeneous baseline (`GCN`). `MP-GNN` does not provide the performance boost that was observed when applied to knowledge graphs (Ferrini et al., 2024), confirming our intuition that existential quantification of meta-path is insufficient when dealing with relational databases. On the other hand, `MPS-GNN` manages to substantially outperform all competitors, thanks to its ability to identify meta-paths that are *informative thanks to the statistics that can be computed over their realizations*, as shown in the following. It is worth noting that this result is achieved with one/two orders of magnitude fewer parameters than the runner-ups, namely `HGN` and `R-HGNN`. See Table 9 in the Appendix for the details. Additionally, Table 10 in the Appendix shows that `MPS-GNN` has competitive execution times with respect to other heterogeneous GNN approaches, thanks to its ability to focus training on relevant meta-path induced subgraphs.

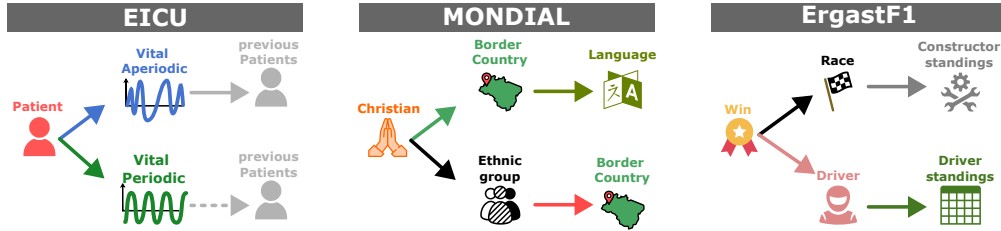

Figure 7: Extracted meta-paths for the three real world datasets.

**Identified Meta-Paths** Figure 7 shows the meta-paths extracted by `MPS-GNN` in the three real world datasets, which clearly convey relevant features for the respective task. For **EICU** (left), meta-paths correlate the patient's length of stay (predictive task) with information on patients with similar periodic (top) and aperiodic (bottom) vital signs. For **MONDIAL** (middle), Christianity is predicted collecting information about the language of border countries (top), and the ethnic group of the country and its neighbouring countries. Finally, in **ErgastF1** the winner is predicted via meta-paths collecting information about the constructor (top) and driver (top) standings.

Finally, in Appendix A.8 we present an experimental evaluation where `MPS-GNN` is adapted to deal with temporal databases and tasks (Robinson et al., 2024), showing how it outperforms its competitors also in this context. Summing up, these results enable us to confidently answer Q2 in the affirmative.

### 5.3 Q3: `MPS-GNN` is a self-explainable method

To address the third research question, we assessed the faithfulness of the extracted meta-paths. The meta-paths identified by the model from the graph are evaluated based on the complementary metrics of sufficiency and necessity. High sufficiency implies that changing the complement to the explanation (leaving the explanation unchanged) should not affect the model's output. High necessity implies that altering the explanation itself (leaving the complement unchanged) should result in a change in the model's output. It is easy to show that our approach is inherently sufficient. Indeed, the computational graph of `MPS-GNN` consists solely of the subgraph containing the occurrences of the identified meta-paths. Necessity, on the other hand, is calculated as a distance metric, measuring the difference in predicted probabilities between the original predictions and those obtained after masking parts of the explanation (i.e. deleting some instances of meta-paths). Defined as $\textsc{Nec} = \frac{1}{N}\sum_{v=1}^{N}(p_v(\mathcal{G}) - p_v(\mathcal{G}'))$ where $v$ is a target node, $\mathcal{G}$ is the original graph and $p_v$ denotes the probability associated with the predicted class. $\mathcal{G}'$ is obtained by removing certain meta-path occurrences (i.e. randomly removing some branches of the identified meta-paths) and $p_v(\mathcal{G}')$ is the probability associated at the class predicted with $p_v$. Since `MPS-GNN` utilizes just the explanation for making prediction one should expect that removing some part of the explanation, has as effect a decrease in prediction $F_1$.

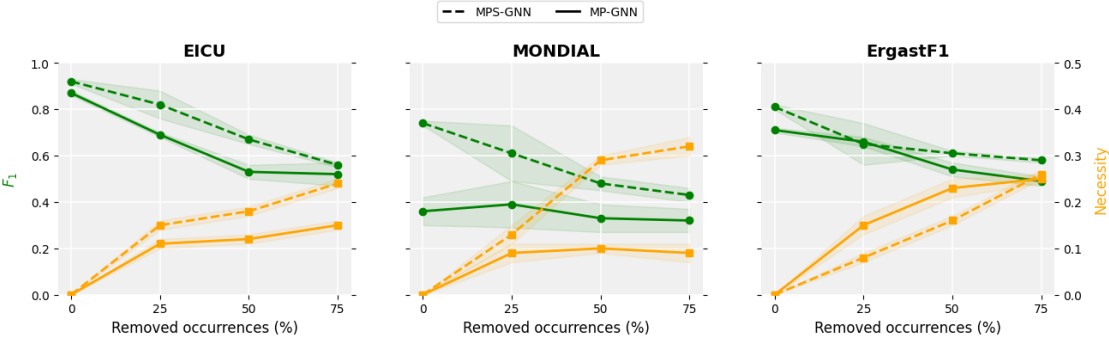

Figure 8: Changes to $F_1$ and posterior probability difference (necessity) when removing 25%, 50%, and 75% of the learned meta-path occurrences for the real-world datasets with `MPS-GNN`, dashed line, and `MP-GNN`, solid line.

Figure 8 illustrates the effects of removing 25%, 50%, and 75% of the meta-path occurrences in terms of changes in $F_1$ and necessity between original and modified graphs (Azzolin et al., 2025). In addition to the results for `MPS-GNN` (dashed line), the figure includes the results for `MP-GNN` (solid line), which is also a self-explainable GNN model according to the reasoning in Section 4.5. In all datasets, there's a noticeable decline in $F_1$ performance for `MPS-GNN` and a steep increase in probability difference, suggesting that the learned meta-paths are also necessary. These results clearly indicate the faithfulness of the explanations of `MPS-GNN`. `MP-GNN` has a similar behaviour, albeit with lower $F_1$ with respect to `MPS-GNN` because of its lower expressivity. The only exception is the **MONDIAL** dataset, where `MP-GNN` fails to learn any relevant pattern, resulting in a very low $F_1$ score that remains approximately constant when removing meta-path occurrences.

## 6 Conclusion

We introduced a novel approach to identify relevant meta-paths of relations for node classification tasks in heterogeneous graphs with a potentially large number of different relations, notably graphs derived from

relational databases. Compared to earlier work, our approach does not require user supervision and learns meta-paths for predictive features defined by aggregate statistics over meta-path occurrences. The explainability of our method is particularly beneficial for sensitive domains like medical or financial data, where it helps address fairness concerns by providing insights into predictions. Experiments demonstrate advantages in accuracy and explainability.

**Limitations and future works**  At this stage, our method is tailored for binary node classification but can be extended to multiclass classification using standard one-vs-all strategies. However, this approach is inefficient, as it requires repeating the entire process for each class. The underlying principles of our local greedy meta-path construction based on scoring potential informativeness of relations directly applies also in the multiclass case. What would be needed in a multiclass adaptation of the approach is to replace the scalar functions (1),(2), and scalar node weights $w_u$, with vector-valued versions, and to modify the loss function (3) accordingly. Similarly, node regression is currently not supported and would require a modification to the way relations are scored. Another limitation of the scoring function is its reliance on a well-connected graph structure; when target nodes have neighbors that are not connected to other target nodes, the scoring function requires a preprocessing step to create supernodes within the neighborhood, as implemented in certain real-world scenarios.
An interesting direction for future work involves incorporating temporal information into the process of learning optimal meta-paths. By doing so, the model would be capable of accounting for the temporal dimension, enabling it to better capture time-dependent relationships and dynamics within the graph.

## 7   Acknowledgments

Funded by the European Union. Views and opinions expressed are however those of the author(s) only and do not necessarily reflect those of the European Union or the European Health and Digital Executive Agency (HaDEA). Neither the European Union nor the granting authority can be held responsible for them. Grant Agreement no. 101120763 - TANGO. AL acknowledges the support of the MUR PNRR project FAIR - Future AI Research (PE00000013) funded by the NextGenerationEU.

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

# A   Appendix

## A.1   Second iteration loss computation

In this Section we show the loss computation and minimization for relation "d" of the example in Figure 5. Note that $Z$ refers to the learned parameters related to the features of the target nodes in the previous iteration with relation $b$ (Figure 3b) and $\begin{bmatrix} 1 \\ 0 \end{bmatrix}$ and $\begin{bmatrix} 0 \\ 1 \end{bmatrix}$ represent the features of *prescription* nodes.

$$Z \gg 0$$

$$F(\mathcal{B}_2, d) = Z\Theta^T \begin{bmatrix} 0 \\ 1 \end{bmatrix} + 2Z\Theta^T \begin{bmatrix} 1 \\ 0 \end{bmatrix} (w_8 + w_9) + Z\Theta^T \begin{bmatrix} 1 \\ 0 \end{bmatrix} (w_{10} + w_{11})$$

$$F(\mathcal{B}_3, d) = 2Z\Theta^T \begin{bmatrix} 1 \\ 0 \end{bmatrix} (w_8 + w_9) + Z\Theta^T \begin{bmatrix} 0 \\ 1 \end{bmatrix} (w_{10} + w_{11}) + Z\Theta^T \begin{bmatrix} 1 \\ 0 \end{bmatrix} (w_{12})$$

$$L(d) = \min_{\Theta, w} \sigma \left( F(\mathcal{B}_3, d) - F(\mathcal{B}_2, d) \right)$$

$$= \min_{\Theta, w} \sigma \left( Z\Theta^T \begin{bmatrix} 1 \\ 0 \end{bmatrix} (2w_8 + 2w_9 + w_{12} - 2w_8 - 2w_9 - w_{10} - w_{11}) \right.$$

$$\left. + Z\Theta^T \begin{bmatrix} 0 \\ 1 \end{bmatrix} (w_{10} + w_{11} - 1) \right)$$

$$= \min_{\Theta, w} \sigma \left( Z\Theta^T \begin{bmatrix} 1 \\ 0 \end{bmatrix} (w_{12} - w_{10} - w_{11}) + Z\Theta^T \begin{bmatrix} 0 \\ 1 \end{bmatrix} (w_{10} + w_{11} - 1) \right)$$

$$\Theta^T \begin{bmatrix} 1 \\ 0 \end{bmatrix} \gg 0; \quad \Theta^T \begin{bmatrix} 0 \\ 1 \end{bmatrix} = 0; \quad w_{12} = 0; \quad w_{10}, w_{11} = 1$$

## A.2   Ablation study

In this section, we highlight the importance of the skip connection $W_1^{(l)} h_v^{(0)}$ considered in equation 7. Table 3 and 4 demonstrate respectively the significance of skip connections in the synthetic and real-world settings. Without considering the initial target node features, the $F_1$ score drops significantly, underscoring the critical role these features play.

Table 3: $F_1$ metric with standard deviations for synthetic datasets with `MPS-GNN` and `MPS-GNN` without using skip connection.

|  | S1 | S2 | S3 | S4 | S5 | S6 | S7 | S8 |
|---|---|---|---|---|---|---|---|---|
| `MPS-GNN` | **0.98**(±0.00) | **0.98**(±0.01) | **0.99**(±0.10) | **0.98**(±0.00) | **0.99**(±0.00) | **0.93**(±0.10) | **0.94**(±0.00) | **0.95**(±0.00) |
| `MPS-GNN`$_{\text{no\_skip}}$ | 0.91(±0.01) | 0.93(±0.01) | 0.89(±0.02) | 0.91(±0.01) | 0.88(±0.00) | 0.87(±0.02) | 0.91(±0.01) | 0.85(±0.03) |

Table 4: $F_1$ metric with standard deviations for real-world datasets with `MPS-GNN` and `MPS-GNN` without using skip connection.

|  | **EICU** | **MONDIAL** | **ErgastF1** |
|---|---|---|---|
| `MPS-GNN` | **0.92**(±0.01) | **0.74**(±0.01) | **0.83**(±0.02) |
| `MPS-GNN`$_{\text{no\_skip}}$ | 0.85(±0.02) | 0.71(±0.01) | 0.80(±0.01) |

## A.3   Real world setting

In our real-world scenario, we utilized three relational databases, which are detailed below. To convert these databases into heterogeneous graphs, we applied transformations to the attribute columns: categorical attributes were transformed using one-hot encoding, and numerical attributes were normalized to the range $[0, 1]$.

To improve connectivity between target nodes, particularly when the transformation from a relational database to a graph results in each target node (or row in the target table) becoming a separate connected component, we employed simple clustering techniques on the rows of other tables based on their features.

Below, we provide detailed information about the databases and describe the clustering methods used when applicable.

**EICU**   Medical database with 31 tables (node types)[4] from Johnson et al. (2021). The task is predicting the duration of a patient's stay in the eICU after admission, modeled as binary node classification by thresholding duration at 20 hours to achieve two balanced classes. To create clusters of nodes, where each cluster is represented by a single new node that replaces all the nodes within that cluster, we utilized a categorical attribute for each table that is best suited for clustering the specific table.

Table 6 provides details about the clustering process applied to the nodes of the **EICU** database. For each table, the initial number of rows and the resulting number of clusters (representing the final number of nodes for that type) are shown. The "Clustering Feature" column specifies the column used for creating clusters; if not specified, this indicates the absence of categorical features, and the DBSCAN algorithm is used instead.

**MONDIAL**   Database [5] containing data from multiple geographical web data sources (May, 1999). We predict the religion of a country as Christian (positive) with 114 instances vs. all other religions with 71 instances. In this dataset, clustering of tables is done using DBSCAN (Ester et al., 1996) clustering algorithm.

Table 7 shows the resulting number of clusters for each table of the original database. Clustering is computed using DBSCAN algorithm.

**ErgastF1**   Database [6] containing Formula 1 races from the 1950 season to the present day. It contains detailed information including lap times, pit stop durations, and qualifying performance for all races up to 2017. The objective is to predict the winner of a race using the data available before the race starts, such as the list of participants and qualifying times, while the actual lap times during the race are not yet available.

Table 5: Setting of real-world datasets. $|\mathcal{T}|$ and $|\mathcal{R}|$ refers respectively to the total number of tables in the original database and the number of relations in the graph used by the models. *Rows* is the sum of all the rows of each specific database.

| Database | $|\mathcal{T}|$ | $|\mathcal{R}|$ | *Rows* |
|---|---|---|---|
| **EICU** | 31 | 87 | 457325320 |
| **MONDIAL** | 40 | 45 | 21497 |
| **ErgastF1** | 14 | 33 | 544056 |

### A.4   Details of competitors' architectures

In Table 8, we present the hyperparameters for all competitors and `MPS-GNN` on the real-world datasets. For the synthetic cases, the only difference lies in the **# layers**, which is adjusted to match the length of the correct meta-paths for the competitors. For the losses **CE** is cross-entropy and **nll** is negative log likelihood.

### A.5   Number of parameters

In Table 9, we present the total number of parameters required for evaluating the various models. In the synthetic setting, when comparing with the only two models that yield satisfactory results, we observe that our approach has a similar number of parameters as `RGCN` (when the total number of relations in the graphs is limited) and `MP-GNN`. `MP-GNN`, which also considers only a subset of graph relations like our

---

[4]https://eicu-crd.mit.edu
[5]https://relational-data.org/dataset/Mondial
[6]https://relational-data.org/dataset/ErgastF1

Table 6: Tables from the **EICU** dataset. 'Clustering Feature' refers to the feature used to group the rows in each table. If not present, it means that the specific table does not have any feature for that purpose, so the DBSCAN algorithm is employed. *Clusters* indicates the final number of nodes after the clustering step.

| Table name | Clustering Feature | *Rows* | *Clusters* |
|---|---|---|---|
| admissiondrug | drughiclseqno | 7417 | 578 |
| admissiondx | admitdxname | 7578 | 268 |
| allergy | drughiclseqno | 2475 | 251 |
| apacheapsvar | - | 2205 | 200 |
| apachepatientresult | - | 3676 | 200 |
| apachepredvar | - | 2205 | 200 |
| careplancareprovider | specialty | 5627 | 40 |
| careplaneol | specialty | 5627 | 40 |
| careplangeneral | cplgroup | 3314 | 28 |
| careplangoal | cplcategory | 3633 | 9 |
| careplaninfectiousdisease | cplcategory | 112 | 11 |
| customlab | labothername | 30 | 19 |
| diagnosis | diagnosisstring | 24978 | 110 |
| hospital | region | 186 | 4 |
| infusiondrug | drugname | 38256 | 257 |
| intakeoutput | celllabel | 100466 | 740 |
| lab | labname | 434660 | 147 |
| medication | drughiclseqno | 75604 | 1027 |
| microlab | organism | 342 | 16 |
| note | notepath | 24758 | 360 |
| nurseassessment | cellattributepath | 91589 | 81 |
| nursecare | cellattributepath | 42080 | 19 |
| nursecharting | nursingchartcelltypevalname | 1477163 | 49 |
| pasthistory | pasthistorypath | 12109 | 190 |
| physicalexam | physicalexampath | 84058 | 310 |
| respiratorycare | currenthistoryseqnum | 5436 | 243 |
| respiratorycharting | respchartvaluelabel | 5436 | 243 |
| treatment | treatmentstring | 38290 | 414 |
| vitalaperiodic | - | 274088 | 200 |
| vitalperiodic | - | 1634960 | 200 |

method, is designed to have a lower number of parameters; however, it still falls short of matching `MPS-GNN`'s performance.

In the real-world setting, among the models that achieve decent results, `GCN` exhibits the lowest number of parameters on the **EICU** dataset. However, among the relational methods, `MPS-GNN` emerges as the most efficient. On the **MONDIAL** dataset, the two leading competitors, `HGN` and `R-HGNN`, despite achieving lower $F_1$ scores, utilize all edge types and consequently require a significantly larger number of parameters.

Finally, on the **ErgastF1** dataset, although `MP-GNN` outperforms other methods in terms of parameter efficiency by considering different meta-paths, it results in a considerably lower $F_1$ score. In contrast, `HGN` and `R-HGNN` exhibit an exponential increase in the number of parameters.

Table 7: Tables from the **MONDIAL** dataset. *Clusters* indicates the final number of clusters after applying DBSCAN algorithm on the features of the specific table.

| Table name | *Rows* | *Clusters* |
|---|---|---|
| economy | 234 | 5 |
| ethnicGroup | 540 | 65 |
| language | 144 | 20 |
| politics | 238 | 25 |
| population | 238 | 4 |
| encompasses | 242 | 2 |
| province | 1450 | 18 |
| organization | 153 | 15 |
| continent | 5 | 5 |
| city | 3111 | 93 |
| river | 218 | 24 |
| sea | 34 | 17 |
| desert | 63 | 6 |
| lake | 130 | 16 |
| mountain | 241 | 40 |

Table 8: Hyperparameters of competitors and `MPS-GNN` for the real-world datasets. The optimizer is omitted from the table as it is Adam for all models. **lr** denotes the learning rate, **wd** represents the weight decay, and **Patience** indicates the early stopping patience (if applicable).

| Hyperparameters | # layers | Embedding dim. | lr | wd | # epochs | Patience | Loss |
|---|---|---|---|---|---|---|---|
| MLP | 2 | 32 | 0.01 | 0.0005 | 500 | 50 | nll |
| GCN | 2 | 32 | 0.01 | 0.0005 | 500 | 50 | nll |
| RGCN | 2 | 32 | 0.01 | 0.0005 | 500 | 50 | nll |
| R-HGNN | 2 | 64 | 0.001 | 0 | 200 | 50 | CE |
| HGN | 2 | 64 | 0.0001 | 0.0005 | 300 | 30 | nll |
| GTN | 2 | 64 | 0.01 | 0.001 | 200 | - | CE |
| Fast-GTN | 2 | 64 | 0.01 | 0.001 | 200 | - | CE |
| MP-GNN | 2 | 32 | 0.01 | 0.0005 | 500 | 50 | nll |
| MPS-GNN | 2 | 32 | 0.01 | 0.0005 | 500 | 50 | nll |

Table 9: Number of parameters for each model across synthetic and real-world datasets.

| | MLP | GCN | RGCN | R-HGNN | HGN | GTN | Fast-GTN | MP-GNN | MPS-GNN |
|---|---|---|---|---|---|---|---|---|---|
| S1 | 194 | 690 | 3730 | 525996 | 10927 | 866 | 126902 | 1346 | 3618 |
| S2 | 194 | 690 | 6770 | 787796 | 42314 | 946 | 126942 | 1346 | 3618 |
| S3 | 194 | 690 | 3730 | 525996 | 42314 | 866 | 126902 | 1346 | 3618 |
| S4 | 194 | 690 | 6770 | 787796 | 74125 | 946 | 126942 | 1346 | 3618 |
| S5 | 194 | 690 | 3730 | 525996 | 74125 | 866 | 126902 | 1346 | 6786 |
| S6 | 194 | 690 | 6770 | 787796 | 74125 | 946 | 126942 | 1346 | 6786 |
| S7 | 194 | 690 | 3730 | 525996 | 74125 | 866 | 126902 | 1346 | 6786 |
| S8 | 194 | 690 | 6770 | 787796 | 74125 | 946 | 126942 | 8834 | 6786 |
| **EICU** | 1346 | 3506 | 400690 | 47496672 | 611785 | 32024 | 110408 | 24898 | 12066 |
| **MONDIAL** | 2144 | 90546 | 3709106 | 88396572 | 1142962 | 180554 | 1539457 | 183138 | 234050 |
| **ErgastF1** | 4356 | 198142 | 5422432 | 396543021 | 439021 | 2542354 | 11325242 | 23413 | 29538 |

## A.6 Execution times

In Table 10, we present the training times for each model across the individual datasets. In the synthetic settings, we observe that among models achieving a significant $F_1$ score (`RGCN`, `MP-GNN`, and `MPS-GNN`),

`MPS-GNN` typically demonstrates the shortest execution time. We would like to highlight that our model is specifically designed to learn meaningful meta-paths in networks with many relation types, whereas the synthetic datasets are limited in the number of relation types. In the **EICU** database, while the `MLP` model achieves the shortest execution time, it performs poorly in terms of $F_1$. Among the models with notable results, `MPS-GNN` exhibits the best execution time. In the **MONDIAL** database, all models have relatively low execution times due to the small graph size, as shown in Table 5. However, `MPS-GNN` still achieves the best $F_1$ score. For the **ErgastF1** dataset, while `MLP` again has the lowest execution time, its final accuracy is poor. In contrast, `MPS-GNN` is comparable to `MP-GNN` and `R-HGNN` in terms of execution time but surpasses them by 12 and 11 percentage points in $F_1$ score, respectively. Overall, our approach is consistently neither the quickest nor the slowest, yet it reliably achieves the highest average F1 score across all settings.

Table 10: Training times, in seconds, for each model across synthetic and real-world datasets.

|  | MLP | GCN | RGCN | R-HGNN | HGN | GTN | Fast-GTN | MP-GNN | MPS-GNN |
|---|---|---|---|---|---|---|---|---|---|
| S1 | 66 | 660 | 1094 | 898 | 363 | 388 | 315 | 234 | 322 |
| S2 | 67 | 660 | 2197 | 674 | 375 | 570 | 812 | 1461 | 245 |
| S3 | 73 | 675 | 536 | 612 | 380 | 260 | 456 | 657 | 356 |
| S4 | 72 | 483 | 2142 | 551 | 373 | 697 | 369 | 986 | 457 |
| S5 | 69 | 420 | 445 | 575 | 360 | 875 | 845 | 158 | 467 |
| S6 | 69 | 620 | 111 | 616 | 377 | 567 | 467 | 453 | 321 |
| S7 | 72 | 677 | 285 | 519 | 371 | 834 | 442 | 587 | 449 |
| S8 | 74 | 777 | 167 | 680 | 373 | 878 | 765 | 1502 | 490 |
| **EICU** | 342 | 5882 | 4355 | 4842 | 3210 | 10324 | 682 | 5787 | 1273 |
| **MONDIAL** | 156 | 125 | 132 | 131 | 265 | 220 | 119 | 120 | 134 |
| **ErgastF1** | 543 | 954 | 1491 | 2456 | 2245 | 3015 | 2945 | 2280 | 2421 |

## A.7 Scoring Function Lookahead Illustration

We report an additional experiment demonstrating the lookahead capabilities implemented in the scoring function of our method, and its advantage over a simplistic greedy approach. We construct a synthetic dataset of a multi-relational graph with three relations and the ground truth metapath, $r_1, r_2$ for the target. We compare our approach against a simple greedy one, in which one always extends the metapath with the relation for which a trained `MPS-GNN` achieves maximal $F_1$ score. The results are presented in Table 11. In the first iteration of the scoring function, relation $r_1$ achieves the lowest loss in our scoring function and would therefore be chosen as the first relation of the metapath. Looking only at the immediate benefit of the relations in terms of the accuracy achieved by a corresponding `MPS-GNN`, however, $r_2$ would be selected as the best relation. The column $r_2$-Extensions shows the $F_1$ scores of all possible length 2 metapaths starting with $r_2$. Comparing with the $F_1$ score of the ground truth metapath we find that, indeed, starting the metapath with $r_2$ is a suboptimal choice, and that our scoring function correctly identifies the most informative relation to start the metapath with, even though this informativeness only is materialized after extension of the metapath with $r_2$.

Table 11: Comparison: our metapath construction vs. simple greedy alternative.

|  | Iteration 1 | | | $r_2$- Extensions | | | Ground truth |
|---|---|---|---|---|---|---|---|
| Meta-paths | $r_1$ | $r_2$ | $r_3$ | $r_2, r_1$ | $r_2, r_2$ | $r_2, r_3$ | $r_1, r_2$ |
| Score | **0.001** | 45 | 56 | | | | |
| $F_1$ | 0.79 | **0.82** | 0.69 | 0.83 | 0.82 | 0.85 | **0.99** |

### A.8 Temporal experiment

Recently, a novel benchmark, rel-bench Robinson et al. (2024), has been introduced. This benchmark consists of multiple relational databases represented as temporal graphs. Additionally, the authors propose `RDL`, a temporal-aware relational message-passing model. It's important to note that the released temporal graphs are not directly compatible with static models. Therefore, in this section, we reconstruct one dataset from the rel-bench repository in a static format and apply our method to this static representation of the graph (setting in table 12). The tasks are: (1) **dnf**, predicting whether a driver will fail to finish a race within the next month, and (2) **top3**, predicting if a driver will place in the top 3. To handle these temporal tasks, we treated each instance of a node at different timestamps as distinct nodes with separate label predictions. Table 13 reports the $F_1$-scores for `MPS-GNN` and the baselines. Our approach outperforms all competitors, including `RDL` which performs lower due to overpredicting the majority class. Note that `RDL` cannot be straightforwardly applied to our other datasets, being designed for temporal datasets. In Section A.8.1, we provide the necessity calculations for these datasets.

Table 12: Setting of temporal real-world dataset.

| Database | $|\mathcal{T}|$ | $|\mathcal{R}|$ | *Rows* |
|---|---|---|---|
| **rel-f1** | 9 | 26 | 97606 |

Table 13: $F_1$-score with standard deviations of our method and competitors on two temporal datasets.

|  | **rel-f1-dnf** | **rel-f1-top3** |
|---|---|---|
| MLP | $0.48(\pm0.00)$ | $0.48(\pm0.00)$ |
| GCN | $0.57(\pm0.01)$ | $0.52(\pm0.02)$ |
| RGCN | $0.44(\pm0.02)$ | $0.54(\pm0.02)$ |
| R-HGNN | $0.60(\pm0.01)$ | $0.63(\pm0.02)$ |
| HGN | $0.61(\pm0.02)$ | $0.61(\pm0.01)$ |
| GTN | $0.41(\pm0.02)$ | $0.45(\pm0.01)$ |
| Fast-GTN | $0.51(\pm0.01)$ | $0.50(\pm0.02)$ |
| MP-GNN | $0.54(\pm0.02)$ | $0.52(\pm0.02)$ |
| RDL | $0.58(\pm0.03)$ | $0.53(\pm0.7)$ |
| MPS-GNN | $\mathbf{0.62}(\pm0.02)$ | $\mathbf{0.65}\ (\pm0.01)$ |

#### A.8.1 Necessity calculation in temporal tasks

In this section, we present the necessity calculation for the temporal datasets. Specifically, as detailed in Section 5.3, Table 14 and figure9 demonstrates that removing certain identified meta-paths results in decreased performance ($F_1$) and an increased necessity value. This finding confirms that the learned meta-paths are essential for the prediction task in these datasets as well.

Table 14: Changes to $F_1$ and necessity when removing 25%, 50%, and 75% of the learned meta-path occurrences for the real-world temporal tasks **rel-f1-dnf** and **rel-f1-top3**.

|  | **$F_1$** |  |  |  | **Necessity** |  |  |  |
|---|---|---|---|---|---|---|---|---|
| Removed (%) | **0** | **25** | **50** | **75** | **0** | **25** | **50** | **75** |
| **rel-f1-dnf** | 0.63 | 0.56 | 0.49 | 0.48 | 0 | 0.12 | 0.22 | 0.31 |
| **rel-f1-top3** | 0.65 | 0.62 | 0.55 | 0.50 | 0 | 0.19 | 0.34 | 0.34 |

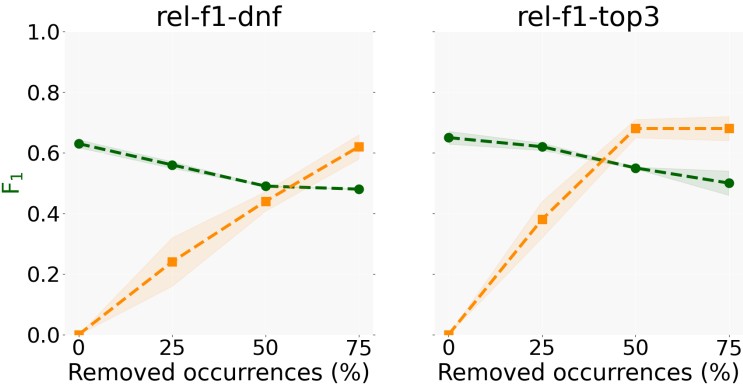

Figure 9: Changes to $F_1$ and posterior probability difference (necessity) when removing 25%, 50%, and 75% of the learned meta-path occurrences for the real-world temporal tasks **rel-f1-dnf** and **rel-f1-top3**.

### A.9 Non-GNN models on real-world databases

For the **MONDIAL** and **ErgastF1** databases, non-GNN methods have shown competitive performance in the past. For example, Schulte et al. (2013); Bina et al. (2013) report $F_1$ scores of 0.78 and 0.77 on **MONDIAL**, 0.4 and 0.3 points higher than `MPS-GNN`. However, these results are achieved on a simplified version of the database with only 12 tables, requiring manual feature selection. In contrast, `MPS-GNN` is applied directly to the raw input data across all 40 tables. The non-GNN methods use Multi-relational Bayes Net Classifiers and Simple Decision Forests, where reducing the number of tables and relations aids performance. By comparison, `MPS-GNN` is designed to handle scenarios with a large number of relations effectively.

### A.10 Toy example with more complex features

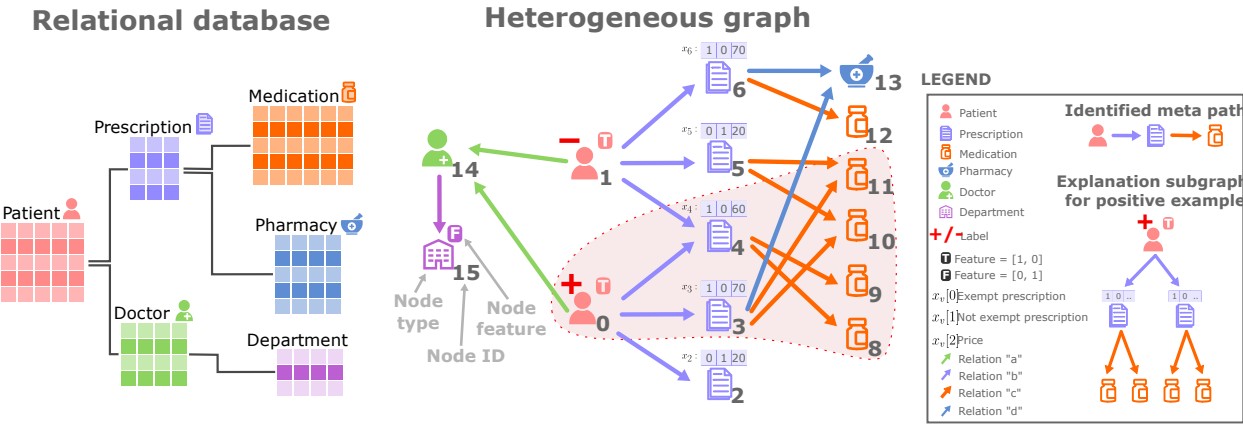

Figure 10: Toy example similar to Figure 1 with more complex features on prescription nodes. As depicted in the legend (**right**), the first two positions of the feature vector determine an exempt prescription ($[1, 0]$) or a not exempt prescription ($[0, 1]$). The last value determine the price of the prescription in dollar. The highlighted subgraph shows a prototypical counts-of-counts pattern characterising positive patients, namely having exempt prescriptions where the total cost is more then 100 dollars, each containing at least two medications

In Figure 10, we present a scenario similar to the one depicted in Figure 1. The key difference lies in the prescription nodes, which now have more complex feature representations. Specifically, the last value of the feature vector corresponds to the price of each prescription.

The first iteration of the scoring function follows the same process as in the previous toy example, as illustrated in Figures 3a and 3b. The most notable aspect, however, is the second iteration, where the target nodes are prescription nodes, similar to Figure 5, where the model evaluates relation $d$ and $c$. Here, we provide the computations involved in minimizing the loss. Notably, only relation $d$ allows the loss to be minimized to 0. Therefore, relation $d$ will be the one included in the identified meta-path

**Minimization relation $c$**

$$Z \gg 0$$

$$F(\mathcal{B}_2, c) = 2Z\Theta^T \begin{bmatrix} 1 \\ 0 \\ 60 \end{bmatrix} + Z\Theta^T \begin{bmatrix} 1 \\ 0 \\ 70 \end{bmatrix} (w_{13}) + Z\Theta^T \begin{bmatrix} 0 \\ 1 \\ 20 \end{bmatrix}$$

$$F(\mathcal{B}_3, c) = 2Z\Theta^T \begin{bmatrix} 1 \\ 0 \\ 60 \end{bmatrix} + Z\Theta^T \begin{bmatrix} 0 \\ 1 \\ 20 \end{bmatrix} + Z\Theta^T \begin{bmatrix} 1 \\ 0 \\ 70 \end{bmatrix} (w_{13})$$

$$L(c) = \min_{\Theta, w} \sigma \left( F(\mathcal{B}_3, d) - F(\mathcal{B}_2, d) \right)$$

$$= \min_{\Theta, w} \sigma (0)$$

$$= \frac{1}{2}$$

**Minimization relation $d$**

$$Z \gg 0$$

$$F(\mathcal{B}_2, d) = 2Z\Theta^T \begin{bmatrix} 1 \\ 0 \\ 60 \end{bmatrix} (w_8 + w_9) + Z\Theta^T \begin{bmatrix} 1 \\ 0 \\ 70 \end{bmatrix} (w_{10} + w_{11}) + Z\Theta^T \begin{bmatrix} 0 \\ 1 \\ 20 \end{bmatrix}$$

$$F(\mathcal{B}_3, d) = 2Z\Theta^T \begin{bmatrix} 1 \\ 0 \\ 60 \end{bmatrix} (w_8 + w_9) + Z\Theta^T \begin{bmatrix} 0 \\ 1 \\ 20 \end{bmatrix} (w_{10} + w_{11}) + Z\Theta^T \begin{bmatrix} 1 \\ 0 \\ 70 \end{bmatrix} (w_{12})$$

$$L(d) = \min_{\Theta, w} \sigma \left( F(\mathcal{B}_3, d) - F(\mathcal{B}_2, d) \right)$$

$$= \min_{\Theta, w} \sigma \left( Z\Theta^T \begin{bmatrix} 1 \\ 0 \\ 70 \end{bmatrix} (w_{12}) + Z\Theta^T \begin{bmatrix} 0 \\ 1 \\ 20 \end{bmatrix} (w_{10} + w_{11}) - Z\Theta^T \begin{bmatrix} 1 \\ 0 \\ 70 \end{bmatrix} (w_{10} + w_{11}) - Z\Theta^T \begin{bmatrix} 0 \\ 1 \\ 20 \end{bmatrix} \right)$$

$$= \min_{\Theta, w} \sigma \left( Z\Theta^T \begin{bmatrix} 1 \\ 0 \\ 70 \end{bmatrix} (w_{12}) + Z\Theta^T \begin{bmatrix} 0 \\ 1 \\ 20 \end{bmatrix} (w_{10} + w_{11} - 1) - Z\Theta^T \begin{bmatrix} 1 \\ 0 \\ 70 \end{bmatrix} (w_{10} + w_{11}) \right)$$

$$\Theta^T \begin{bmatrix} 0 \\ 1 \\ 20 \end{bmatrix} \gg 0; \quad w_{10}, w_{11}, w_{12} = 0$$

