# OpenReview forum: "A Self-Explainable Heterogeneous GNN for Relational Deep Learning"
_TMLR — Accepted by TMLR_

### Review · Reviewer_7hSS · 2024-12-18

**Summary Of Contributions:**

The paper proposes a method for applying GNNs to relational databases, claiming state-of-the-art performance. Experiments include various datasets (synthetic and real-world) and numerous baselines.

**Audience:**

Yes

**Broader Impact Concerns:**

some acknowledgement of model Fairness concerns when working with sensitive data (medical, financial etc.) might be added to the Discussion section

**Claims And Evidence:**

Yes

**Requested Changes:**

- I suggest providing a more detailed toy problem example (similar to MP-GNN paper), in which the meta-path construction, and the GNN role are clearly explained step-by-step.
Also, in the Introduction I'd prefer to have more context on the tasks the proposed approach can be applied to. (In my opinion, Figure 1 alone is not enough). Other examples than the count-on-count prediction might also help further.

- adding runtime complexity analysis and/or discuss the runtime scalability of MPS-GNN to as the datasets increase (in terms of databases sizes, #relations, #features).

- adding more details about the baselines

- adding the Discussion section

**Strengths And Weaknesses:**

Strengths:
- MPS-GNN shows better results than the considered baselines
- the authors focus on explainability of the developed method
- the paper is well structured, with articulated research questions and answers to them
- Experiments include real-world datasets

Weaknesses:
- Not enough information about the baselines: how their hyperparameters were tuned and which architectures used exactly? (for example, there are no details about the MLP baseline architecture )
- Discussion is missing. Potential points to discuss in it: limitations of MPS-GNN; potential for extensions (e.g., regression problems);  future work

- The authors provide the runtimes of MPS-GNN and the baselines, but it would be good to have some time complexity analysis to understand how well MPS-GNN scales.

---

> ### Author Response · Authors · 2025-01-09
>
> **Figures modification**
>
> We have revised Figure 1 to make it clearer. Additionally, we split the old Figure 2 into four separate figures to explain, step by step and in greater detail, what happens each time the score of a relationship is computed. As a result, Figures 2 and 3 now refer to the first iteration of the scoring function, while Figures 4 and 5 correspond to the second iteration.
>
> **Details of competitors’ architecture**
>
> In Appendix A.4 we added a description of the competitors’ architecture with a Table describing the implementation details for all of them.
>
> **Missing discussion**
>
> We expanded the Conclusion section as suggested, discussing potential limitations of the current solution and suggesting possible directions for future work.
>
> **Complexity analysis**
>
> In Section 4.2.1, we included a complexity analysis of the scoring function, showing how it allows to reduce the complexity of the meta-path search procedure from polynomial to linear in the number of candidate relations.

---

### Review · Reviewer_GSzS · 2024-12-23

**Summary Of Contributions:**

This work proposes a self-explainable heterogeneous graph-based approach for relational learning. The method leverages aggregated statistics of meta-path co-occurrences to learn meta-paths for predictive tasks, (i.e., the binary classification of graph nodes). The proposed approach outperforms existing approahces in identifying relevant meta-paths for predictive tasks on both synthetic and real-world relational datasets. The faithfulness of the learned meta-paths is validated through controlled experiments by selectively removing a certain proportion of learned meta-path occurrences.

**Audience:**

Yes

**Claims And Evidence:**

Yes

**Requested Changes:**

Critical for acceptance:
- I would recommend a more in-depth discussion of the MP-GNN baseline and the proposed approach regarding the RQ on the self-explainable aspect of the method.

**Strengths And Weaknesses:**

Strengths:
- The paper is overall well-written.
- Annonomous code is available.
- This work conducts multiple experiments to address the three research questions (RQs) and provides strong empirical evidence demonstrating that the proposed method 1. can recover the correct meta-paths with increasing complexity 2. outperforms existing approaches on real-world relational datasets 3. is a self-explainable method.

Weaknesses:
- It appears that the previous approach, MP-GNN [1], serves as the fundamental baseline for the proposed method, as also outlined in Section 4.6. However, if I am not missing the information, in Section 5.3, the same F1 and necessity tests (Table 3 and Figure 5) are not conducted or presented for the MP-GNN baseline.

- It would be helpful to include a visualization of the synthetic data, if possible, or a figure demonstrating the increasing complexity of synthetic data from S1 to S8 as the number of relations increases, apart from figure 3.


[1] Francesco Ferrini, Antonio Longa, Andrea Passerini, and Manfred Jaeger. Meta-path learning for multirelational graph neural networks. In Soledad Villar and Benjamin Chamberlain (eds.), Proceedings of the Second Learning on Graphs Conference, volume 231 of Proceedings of Machine Learning Research, pp. 2:1–2:17. PMLR, 27–30 Nov 2024. URL https://proceedings.mlr.press/v231/ferrini24a.html.

---

> ### Author Response · Authors · 2025-01-09
>
> **Self-explainability analysis of MP-GNN**
>
> We added the results for MPGNN to Figure 8 (we removed Table 3 to avoid redundancy of information). Results show that, as expected given that it relies on meta-paths, MPGNN is also a self-explainable GNN, albeit a less accurate one with respect to MPS-GNN.
>
> **Synthetic data visualization**
>
> Thanks for the suggestion. In the table of Figure 5, we now provide the explanation subgraph for each synthetic dataset, illustrating the increasing complexity of the datasets.

---

### Review · Reviewer_tkeZ · 2025-01-01

**Summary Of Contributions:**

The paper introduces Meta-Path Statistics GNN (MPS-GNN) which builds on ideas of MP-GNN and learns meta-path based on learnable statistics. The MPS-GNN can be particularly useful for learning over relational databases. The learned meta-paths can further server as model-level explanations making the model self-explainable. The paper introduces adaptations to the MP-GNN algorithms and tests the method on multiple synthetic and real datasets in order to order to answer three research questions related to i) recovery of the correct meta-path, ii) performance in comparison to existing methods, and iii) self-explainability and answers all three favorably.

**Audience:**

Yes

**Claims And Evidence:**

No

**Requested Changes:**

1. The organization of the text in the Methodology section is very unclear even confusing, and makes understanding of the method very difficult. E.g.
- The differences compared to MP-GNN need to be explicitly and much more clearly explained.
- Use clear references to other parts of texts where you explain further details (e.g. from intro of section 4 to later subsections).
- Top of page 5 - you say feature function (2) can be determined by own attributes, putative features $\mathbf{w}$ or a combination of both. What do you mean? The equation clearly has all of these so is determined by all. Also why do you call $\mathbf{w}$ *putative features*? What do you mean by this?
- What is the motivation for the loss-function in equation (3)? What is the motivation for the sigmoid transformation (wouldn't the min be at the same param values without the sigmoid)?
- under equation (3) - you say in practice you approximate by sampling the bags. Why? The equation is already written as a sum over bags in the training set, not as an expectation over population nodes or similar.
- under equation (4) - you say the meta-path construction can be terminated if $\sum w_u$ term plays "no significant role". What does "significant" mean here? This has many standard interpretations in statistics and data science which you probably do not have in mind. Please be clear and precise.
- under equation (4) - you say, you could possibly learn $\mathbf{w}$ by regression. How? Where would the supervision come from?
- Figure 2 bottom - you claim that $alpha(4, \mathcal{B}_2)=alpha(4, \mathcal{B}_3)=2Z$. This is very confusing when comparing to equation (6) which seem to accumulate only across nodes within the same bag. Also first para in section 4.1 says that weights of a node can be different for different bags. Either the notation is confusing or this is not correct.
- Where is equation (7) used? How are the $W$ set? I presumed these are trained? When? Missing in Algo 1.
- Which of the equations is actually the "scoring function"?
- Section 4.4. - how are multiple meta-path learned? The greedy, local approach does not have a unique solution leading to a single best meta-path?
- page 9 top - what do you mean by "existential quantification assumption"?
- What are the last two columns in Table 2?
2. The mathematical description needs complete revamp - there are very many typos/errors and the notation is unclear and confusing. E.g.
- Page 4 2nd para - why do you introduce $u$ as entity and $v$ as node? These are the same or not? Moreover, you use both $u$ and $v$ for nodes later on. Similarly, why introducing $a_u$ and $x_v$ as two notational approaches for the same thing?
- Page 4 para about *Greedy* - indexing of relation re-uses $i+1$ index twice.
- Symbol $F$ is used twice in equation (1), each time meaning something else. It is further used to indicate $F_1$ score. This is extremely confusing.
- It is unclear what the $\cdot$ \cdot symbol in the presented equations means. Is this to be element-wise product or inner product?
- It is unclear what is the dimensionality of the inputs and outputs of the functions - some seem to be scalars, some vectors, see also above point.
- It is unclear if there are some limits on the values of the weights $\alpha$ and attributes. For example in page 5 under equation (6) you claim that nodes have *bigger* weights in subsequent tasks. This would assume that $\alpha$ as well as the node features $x_v$ are always positive, or am I missing something?
- I assume $\theta$ and $\mathbf{w}$ parameters are shared across different $r$
- Links between equations are very unclear - for example, I think $\alpha(v, \mathcal{B})$ and $\mathbf{w}$ are probably somehow related. I'm, however, not sure and definitely not sure how.
- Figure 2 iteration c - is wrong, should be 10 and 11 set to 1 (not 8 and 9).
- This list is probably not exhaustive. Please review and update your math carefully.
- Equation (7) - are these inner products? The dimensions of feature embeddings are the same as the initial feature vector $x_v$? Irrespective of the original Table in the relational database of the individual elements (nodes)?
3. Some claims are unclear, not convincing or not evidenced. Eg.
- All your illustrative examples are on very simplistic problems with no node attributes (simplified to true / false). It is unclear to me how this would generalized to nodes with more complex features.
- Bottom of page 6 - you say can be generalized to multiple meta-path by concatenation. Not clear to me how exactly.
- section 5.1 - the claim in the title is formulated rather generic while the synthetic experiments focus on one problem setting only (at least c occurrences means positive). It is not clear if the method would behave similarly for other settings. Either provide more convincing evidence or reformulate your claims more conservatively.
- Top page 9 - you say look ahead capabilities of scoring function are crucial but in intro of section 4 you say your strategy is greedy and local. Very confusing. Please clarify.
- in section 5.2 - your real experiments in the main text are described as regression or multi-class. It is only apparent from appendix that you binarize the outcomes. Make this clear in the main text.
- Section 5.2 Results - you claim that MPS-GNN outperforms other methods "thanks to its ability to identify meta-path that are informative thanks to the statistics ... ". From the presented results in Table 2 MPS-GNN outperforms other methods but the reason for this better performance is not quite clear.
- Section 5.2 - the title claims self-explainability of the method. The sufficiency and necessity of the meta-path are interesting but in my view do not support a fairly generic claim of self-explainability of the method. It is clear that the path matters for the decision but what are the particular features in the path which make the algorithm to decide 0 or 1?
- The results presented in Table 3 are difficult to interpret. What do you mean by dropping x% of meta-path occurrences? Have these been dropped randomly? Wouldn't changing the explanation mean changing the features within the meta-path rather than dropping meta-paths?
- In conclusions you claim your method offers advantages over other methods in terms of sparsity. I haven't seen any evidence for this.
- This list is probably not exhaustive. Please review your claims carefully and provide convincing evidence or re-formulate.

**Strengths And Weaknesses:**

**Strengths** - learning over relational databases is interesting problem

**Weaknesses** - unfortunately, in my view there are many major ones:
- The paper is technically very badly written, at several places very unclear, failing to present the algorithm accurately.
- Due to insufficient technical soundness of the presentation the presented conclusions are far from convincing.
- Some of the claims of the paper are not supported by any evidence or not sufficiently convincing evidence.

See "Requested Changes" for further details.

---

> ### Author Response · Authors · 2025-01-09
>
> **Organization of Methodology section**
>
> - The differences compared to MP-GNN need to be explicitly and much more clearly explained.
>
> MP-GNN and MPS-GNN, differ fundamentally in their approach to handling meta-path information. While MP-GNN uses a max aggregation strategy, focusing on the existence of a meta-path, MPS-GNN employs a sum aggregation strategy that allows it to count and utilize statistical measures related to meta-path occurrences. Furthermore, MP-GNN relies on additional prediction steps to form positive bags, ensuring that at least one neighbor aligns with the correct meta-path. In contrast, MPS-GNN directly includes all relevant neighbors contributing to multiple meta-path occurrences without requiring such additional steps, making it more effective in relational database contexts where class labels depend on statistical reasoning over meta-paths.
> To clarify these differences, we revised the sub section, “Comparison with MP-GNN,” which explicitly highlights the methodological and practical distinctions between the two approaches. Additionally, we updated the introduction to briefly describe these limitations of MP-GNN in relation to our method.
>
> - Use clear references to other parts of texts where you explain further details (e.g. from intro of section 4 to later subsections).
>
> We referenced in the introduction of section 4 the next subsections.
>
> - Top of page 5 - you say feature function (2) can be determined by own attributes, putative features or a combination of both. What do you mean? The equation clearly has all of these so is determined by all. Also why do you call putative features? What do you mean by this?
>
> We call the learned node weights ‘putative features’ because these values are enabling us to (approximately) solve our prediction task in a GNN architecture. However, they are only ‘putative’ because we do not know (yet) whether these weights can be computed from the data. Subsequent extensions of the meta path are intended to make data accessible from which the weights can be (approximately) computed, i.e. the ‘putative’ features be materialized into actual features. The node feature function in Eq. 2 is indeed computed as a combination of the putative features and the attributes of the node itself. The latter can be made irrelevant if needed, provided they include at least one one-hot encoded categorical attribute (the node type, corresponding to the table name in the relational database). It is sufficient that theta is set to 1 for all entries corresponding to the one-hot encoding, and zero everywhere else. Concerning putative features, when the r-neighborhood is non-informative because neither the number nor the identity of neighbours has discriminative value, then the putative features can be made irrelevant by learning constant weights.
>
> - What is the motivation for the loss-function in equation (3)? What is the motivation for the sigmoid transformation (wouldn't the min be at the same param values without the sigmoid)?
>
> The motivation for the loss function is to serve as a soft approximation for the categorical constraints F(B+)>F(B-) for all pairs B+/B- of positive/negative bags. The sigmoid function ensures that the loss stays bounded. Otherwise, the loss could easily go to -\infty simply by finding parameters \Theta,w and a pair B+,B- for which F(B+,r,\Theta,w) grows unbounded (and F(B-,r,\Theta,w) stays bounded). Since the sigmoid is inside the sum, it is not the case that the same parameters minimize the function with or without the sigmoid (that would only hold if the sigmoid was outside the sum).
>
> - under equation (3) - you say in practice you approximate by sampling the bags. Why? The equation is already written as a sum over bags in the training set, not as an expectation over population nodes or similar.
>
> The number of terms in the sum of (3) is quadratic in the number of training examples. This makes a full evaluation of the sum infeasible in most cases, and therefore we approximate by a random sample of terms.
>
> - under equation (4) - you say the meta-path construction can be terminated if term plays "no significant role". What does "significant" mean here? This has many standard interpretations in statistics and data science which you probably do not have in mind. Please be clear and precise.
>
> We agree that “significant” is a bit vague in this context, and the explanation was misleading. We revised the explanation of the stopping criterion as follows: “If all candidate relations fail to minimize the loss, i.e., optimizing Θ, w does not lead to substantially lower loss than using random parameters (i.e., does not improve by at least 30%), then the meta-path construction terminates and the current meta-path is returned.”

---

> > ### Author Response · Authors · 2025-01-09
> >
> > - under equation (4) - you say, you could possibly learn by regression. How? Where would the supervision come from?
> >
> > The supervision would consist of setting the learned node weights w_u as regression target values (and then using any standard regression loss function).
> >
> > - Figure 2 bottom - you claim that alpha(4,B_2)=alpha(4,B_3)=2Z. This is very confusing when comparing to equation (6) which seem to accumulate only across nodes within the same bag. Also first paragraph in section 4.1 says that weights of a node can be different for different bags. Either the notation is confusing or this is not correct.
> >
> > Yes, weights for the same node can be different in different bags. It only happens to be the case in this example that alpha(4,B_2) is equal to alpha(4,B_3). We modified the text and Figure 3 that now clearly  illustrates that the same node  can have varying alpha values depending on the bag they belong to.
> >
> > - Where is equation (7) used? How are the W set? I presumed these are trained? When? Missing in Algo 1.
> >
> > The equation defines the forward step of the GNN built over the meta-path. Weights are trained with standard supervised GNN training. We made this explicit in Algorithm 1, by separating the GNN training and GNN evaluation steps.
> >
> > - Which of the equations is actually the "scoring function"?
> >
> > The scoring function is the loss function in eq. 4. The relation with minimal loss is chosen for further extension.
> >
> > - Section 4.4. - how are multiple meta-path learned? The greedy, local approach does not have a unique solution leading to a single best meta-path?
> >
> > Yes, a pure execution of the greedy local search will only lead to a single meta path solution. However, this is not guaranteed to be the optimal among all single meta-paths. We therefore use a standard beam search approach to pursue multiple high-scoring (not only the optimal) meta path prefixes. Moreover, we can use several of the discovered meta paths in the MPS-GNN.  We clarified this at the end of section 4.4.
> >
> > - page 9 top - what do you mean by "existential quantification assumption"?
> >
> > The existential quantification assumption we refer to is the one discussed in the Introduction section regarding the MP-GNN method. Specifically, MP-GNN assumes that a node's class label depends on a single occurrence of a meta-path, meaning just one instance is sufficient for accurate classification. While this assumption may appear reasonable for knowledge graphs, it is less clear how applicable it is to relational databases, as demonstrated in our experimental setup. We further clarified this assumption in the Introduction and showed how it is inappropriate for the running example in Figure 1.
> >
> > - What are the last two columns in Table 2?
> >
> > There was an error in including the last two columns of the table, which belong to an additional experiment discussed in Appendix A.8. We thank the reviewer for bringing this to our attention.
> >
> > **Typos in mathematical description**
> >
> > - Page 4 2nd paragraph why do you introduce u as entity and v as node? These are the same or not? Moreover, you use both u and v for nodes later on. Similarly, why introducing a_u and x_v as two notational approaches for the same thing?
> >
> > In the Preliminaries section, we illustrate how to represent a relational database as a graph, using distinct notations for corresponding elements. Specifically, $u$ denotes an entity in the relational database, which, when represented as a node in a heterogeneous graph, is assigned a different notation, $v$. Similarly, attributes are denoted as $a_u$ in the relational database and as $x_v$ in the corresponding heterogeneous graph. There is indeed a notation clash later on in the paper, where we use $u$ to denote nodes in the context of the heterogeneous graph. To avoid confusion, we replaced the notation referred to entity $u$ in relational databases with $e$.
> >
> > - Page 4 paragraph about Greedy - indexing of relation re-uses $i+1$ index twice.
> >
> > Correct, the second mention was meant to be $i+2$. We fixed it.
> >
> > - Symbol F is used twice in equation (1), each time meaning something else. It is further used to indicate $F_1$ score. This is extremely confusing.
> >
> > Thanks for pointing out, we revised the notation using uppercase F for bags and lowercase f for single nodes.
> >
> > - It is unclear what the \cdot symbol in the presented equations means. Is this to be element-wise product or inner product?
> >
> > We used it as an inner product, but we agree that using it also for scalars can be confusing. For better consistency, we revised the notation to get rid of the \cdot symbol altogether.
> >
> > - It is unclear what is the dimensionality of the inputs and outputs of the functions - some seem to be scalars, some vectors, see also above point.
> >
> > All functions have scalar outputs, this should be clearer with the revised formulation.

---

> > > ### Author Response · Authors · 2025-01-09
> > >
> > > - It is unclear if there are some limits on the values of the weights \alpha and attributes. For example in page 5 under equation (6) you claim that nodes have bigger weights in subsequent tasks. This would assume that \alpha as well as the node features x_v are always positive, or am I missing something?
> > >
> > > Admittedly, the discussion of the relevant definitions was not very detailed, and the wording on p.5 misleading. We have now expanded the text above Eq. (1) and below Eq. (6) a little bit in order to clarify: weights (and node attributes) can be both positive or negative. “Bigger” weights have to be understood as larger in absolute value.
> > >
> > > -  I assume \theta and w parameters are shared across different r.
> > >
> > > That’s not the case. \Theta and w are the minimizers of Equation (4), and are thus specific to the relation r that defines the loss function L(r,\Theta,w).
> > >
> > > - Links between equations are very unclear - for example, I think \alpha(w,B) and w are probably somehow related. I'm, however, not sure and definitely not sure how.
> > >
> > > The \alpha(w,B) and w are only very loosely related: they both depend on the learned parameters \Theta. However, it is intentional that the learned weights w do not directly determine the weights \alpha: as explained in the paragraph between equations (4) and (5), there can be many different solutions for w that all minimize (4). The \Theta parameters, on the other hand, typically have fewer degrees of freedom in the minimization of (4) (partly because there will usually be much fewer \Theta than w parameters), and therefore they provide useful information for setting up the weighted multi-instance problem for the next iteration.
> > >
> > > - Figure 2 iteration c - is wrong, should be 10 and 11 set to 1 (not 8 and 9).
> > >
> > > We thanks the reviewer for pointing this out. We modified the figures and we corrected the mistake
> > >
> > > - Equation (7) - are these inner products? The dimensions of feature embeddings are the same as the initial feature vector Irrespective of the original Table in the relational database of the individual elements (nodes)?
> > >
> > > In heterogeneous graphs, GNNs handle different feature dimensions across node types (or tables in relational databases) by using separate layers for each type, which are then mapped to a common embedding space. This is a typical approach in the literature of heterogeneous graph learning.
> > >
> > > **Not convincing claims**
> > >
> > > - All your illustrative examples are on very simplistic problems with no node attributes (simplified to true / false). It is unclear to me how this would generalized to nodes with more complex features.
> > >
> > > In order to facilitate presentation of the running examples we only considered a single Boolean node attribute in the examples. However, the method in general is in no way restricted to this simplistic case (and in the experiments of Sec. 5.2 we deal with datasets with more complex node features).
> > >
> > > - Bottom of page 6 - you say can be generalized to multiple meta-path by concatenation. Not clear to me how exactly.
> > >
> > > To simplify the explanation of the methodology and algorithm, we showed the meta-path construction as selecting only the relationship with the highest score from the scoring function. However, to obtain multiple meta-paths, we can select the top K relationships and build each meta-path independently. The neural network then generates embeddings for the target nodes from each meta-path, with the final representation being their concatenation. We added an explanation of this at the end of section 4.3.
> > >
> > > - section 5.1 - the claim in the title is formulated rather generic while the synthetic experiments focus on one problem setting only (at least c occurrences means positive). It is not clear if the method would behave similarly for other settings. Either provide more convincing evidence or reformulate your claims more conservatively.
> > >
> > > We revised the title as “Q1: MPS-GNN consistently identifies the correct meta-path in count based synthetic scenarios”
> > >
> > > - Top page 9 - you say look ahead capabilities of scoring function are crucial but in intro of section 4 you say your strategy is greedy and local. Very confusing. Please clarify.
> > >
> > > Our approach is greedy only in the sense that it adds one relation at a time, without explicit consideration of the combinatorial space of further metapath extensions. However, the scoring function is specifically defined to identify relations that are **potentially** informative, which is a form of lookahead.
> > >
> > > - in section 5.2 - your real experiments in the main text are described as regression or multi-class. It is only apparent from appendix that you binarize the outcomes. Make this clear in the main text.
> > >
> > > We expanded the explanation of the datasets in the main text mentioning the binarization procedure.

---

> > > > ### Author Response · Authors · 2025-01-09
> > > >
> > > > - Section 5.2 Results - you claim that MPS-GNN outperforms other methods "thanks to its ability to identify meta-path that are informative thanks to the statistics ... ". From the presented results in Table 2 MPS-GNN outperforms other methods but the reason for this better performance is not quite clear.
> > > >
> > > > We have a paragraph on identified meta-paths showing how the meta-paths extracted by the approach are interpretable and intuitively informative for the task at hand. Also the advantage over plain MP-GNN confirms the role of the counting feature of MPS-GNN for achieving the result.
> > > >
> > > > - Section 5.2 - the title claims self-explainability of the method. The sufficiency and necessity of the meta-path are interesting but in my view do not support a fairly generic claim of self-explainability of the method. It is clear that the path matters for the decision but what are the particular features in the path which make the algorithm to decide 0 or 1?
> > > >
> > > > We revised the section to clarify that our claim comes from the notion of self-explainable GNNs, which rely on a detector extracting an “explanation” subgraph and a classifier using it to output the prediction. We also added a discussion on the relevance of feature-level explainability (on which we agree) and on its challenge for existing layerwise node embedding computation architectures.
> > > >
> > > > - The results presented in Table 3 are difficult to interpret. What do you mean by dropping x% of meta-path occurrences? Have these been dropped randomly? Wouldn't changing the explanation mean changing the features within the meta-path rather than dropping meta-paths?
> > > >
> > > > Dropping x% of the meta-path occurrences involves randomly removing segments of the identified meta-paths. As a result, the features of the nodes corresponding to the removed segments are also eliminated, since they are directly part of those meta-paths. We tried to make it more clear in the text.
> > > >
> > > > - In conclusions you claim your method offers advantages over other methods in terms of sparsity. I haven't seen any evidence for this.
> > > >
> > > > When we say that our method demonstrates sparsity compared to other approaches, we are referring to its ability to use only the learned meta-paths for making final predictions. By doing so, we reduce the size of the graph in terms of both nodes and relations, as only those nodes and relations that are part of the identified meta-paths are utilized. Indeed as shown in table 10, the number of parameters is lower with respect to the majority of other methods.

---

> > > > > ### Comment · Reviewer_tkeZ · 2025-01-30
> > > > > **Responses to Not convincing claims**
> > > > >
> > > > > * illustrative examples too simplistic - it is probably clear to you that it readily extends beyond the simple binary true/false case. But I do not find this so obvious. Extending more clearly what would change in the non-binary case or extending the examples to at least one non-binary case would be helpful.
> > > > > * multiple meta-path concatenation - ok
> > > > > * claim in section 5.1 - ok
> > > > > * greedy vs look ahead - ok
> > > > > * binarization - ok
> > > > > * 5.2 - ok
> > > > > * 5.3 explainability - ok
> > > > > * dropping x% - ok
> > > > > * sparsity - I still do not see any evidence for this. MP-GNN is also sparse. Also table 10 compares the training times which is not directly related to sparsity. Did you mean table 9? Even there the number of parameters is not lower than MP-GNN. So I do not see the advantages over MP-GNN in terms of sparsity.

---

> > > > > > ### Author Response · Authors · 2025-02-06
> > > > > >
> > > > > > We sincerely thank the reviewer for this discussion, which is helping us improve our work. Below, we address the points raised.
> > > > > >
> > > > > > **Hyperparameter**
> > > > > >
> > > > > > The 30\% improvement threshold is a hyperparameter chosen empirically. However, the model is robust to changes to this hyperparamer, with the same meta-paths being learned for a range of threshold values. We have updated the algorithm section to account for these insights.
> > > > > >
> > > > > >
> > > > > > **Putative features and w as regression**
> > > > > >
> > > > > > We believe that the questions regarding putative features and w as a regression are indeed related, so we address them together.
> > > > > > We apologize for not being sufficiently clear on this point. To clarify our approach, we have added a new figure (Fig. 2) along with an explanatory text in the manuscript. In particular, the figure and text illustrate the core principle of our method. Given an initial (binary) node classification task, a first relation is identified that could solve the task with the help of a putative node feature (weight) w on the successor nodes. Then a surrogate task is set up whose target is to materialize the putative feature as a feature computable from the data. Instead of treating this task as a regression problem where the weights w, learnt in the previous iteration, become the targets, we reformulate it as a weighted multi-instance classification task.
> > > > > >
> > > > > > **scoring function**
> > > > > >
> > > > > > We clarified in the text in section 4.2
> > > > > >
> > > > > > **existential quantification assumption**
> > > > > >
> > > > > > We have clarified in the Introduction what we mean by the "existential quantification assumption."
> > > > > >
> > > > > > **eq 7 dimensions**
> > > > > >
> > > > > > We mentioned the embedding step of MPS-GNN in section 4.4.
> > > > > >
> > > > > > **illustrative examples too simplistic**
> > > > > >
> > > > > > We added Section A.10 in the Appendix, presenting an example similar to the toy example in Figure 1 but with more complex features in one of the node types. This shows that our scoring function can still correctly identify the relevant relation even in this more challenging scenario.
> > > > > >
> > > > > > **sparsity**
> > > > > >
> > > > > > We apologize for the confusion the term might have produced. When we mentioned that our model offers advantages in terms of sparsity, we were referring to the fact that the explanation subgraph used for final predictions is composed only of the relations and nodes that appear in the learned meta-paths. Since MP-GNN is also based on meta-paths, it shares this sparsity characteristic. However, our model is more expressive due to the absence of strong initial assumptions and achieves higher predictive accuracy.
> > > > > > The primary focus of our approach is to provide a self-explainable model with superior accuracy compared to competitors, while maintaining a relatively low number of parameters.
> > > > > > To avoid potential misinterpretations, we have removed the reference to sparsity in the conclusion, as it did not accurately reflect the main objective of our method.

---

> > > ### Comment · Reviewer_tkeZ · 2025-01-30
> > > **Responses to Typos in mathematical description**
> > >
> > > * u, v, e - ok
> > > * greedy indexing - ok
> > > * F, f - ok
> > > * inner products - ok
> > > * dimensionality of inputs / outputs - ok
> > > * $\alpha$ weights - yes, better, thanks
> > > * $\theta$, w specific for r - or
> > > * w vs $\alpha$ - ok
> > > * figure 2 corection - ok
> > > * eq7 dimensions - is the embedding step mentioned somewhere? sorry, I didn't see it. it should.

---

> ### Comment · Reviewer_tkeZ · 2025-01-30
> **Responses to Organization of Methodology section (sequentially)**
>
> * compairosn to MP-GNN - yes, thanks, better
> * refs- ok
> * comments to eq(2) - yes, thanks, better
> * clarifications for eq(3) -ok
> * eq(4) comments - more clear now but the provided threshold of 30% seems to rather arbitrary. Is this a hyper-parameter to be set by the user? As such, I reckon it should be included into Algo 1? Also I do not understand what you mean by "the problem becomes capturing the putative node features w by actual features." Do you mean this is the only problem that remains to be solved? What do you mean by "capturing" putative features by actual features?
> * w as regression - sorry, I do not get it. isn't the w learned as a parameter? How could you use it as the target in the regression problem? Are these not unknown and hence unavailable as targets? I am probably missing something.
> * Fig2, Fig3 and comments - yes, much better, thanks
> * eq(7) - ok
> * scoring function - please make this clear in the text
> * multiple meta-path - yes, better, thanks
> * existential quantification assumption - please make clear in the Intro or somewhere else appropriate that you refer to the concept by this specific term thereafter in the text
> * last 2 columns table 2 - ok

---

### Author Response · Authors · 2025-01-09

We thank the reviewers for taking their time to review our paper. We have substantially revised the paper taking into account the reviewers’ feedback. Among the major changes (highlighted in blue), we have rewritten certain sections of the paper to enhance coherence and ensure mathematical accuracy. Additionally, we have revised several figures to improve their clarity. Below, we provide detailed responses to the specific requests and comments made by each reviewer.

---

### Decision · Action_Editor_Abxk · 2025-02-19

**Recommendation:** Accept as is

**Comment:**

We received three reviews, two of which were positive and one (tkeZ) negative. The first two reviewers highlighted a few concerns related mostly to the discussion of the method (hyper-parameters choice, runtime complexity, visualization). They did not engage further in the rebuttal, but in my opinion the author's answers were satisfying on all counts. Reviewer tkeZ had a very broad set of (valid) concerns related to the clarity of the method, the notation, and the organization, which were addressed in a complete way during rebuttal (no point remains open). Overall, all reviewers agree on the validity of the approach and on the experimental evaluation.

**Audience:**

The paper targets explicitly what they call "relational deep learning", which is the application of GNN models to relational databases. However, the method is general for broader classes of heterogeneous graphs (with limitations clearly described in the paper), making it a potentially valid contribution to the entire geometric DL field.

**Claims And Evidence:**

The paper describes an algorithm for node classification in heterogeneous graphs built from relational databases. The key idea is to identify paths in the graph (called meta-paths) in a greedy fashion, which are informative w.r.t. the node labels. The approach extends a previous paper (MP-GNN) by replacing an existential quantifier on the meta-paths (given by a max operation) with a more general sum operation that allows the model to operate on partial statistics (i.e., multiple occurrences) of meta-paths.

The extension, while small, is well motivated in the introduction thanks to very clear illustrations and toy visualizations. The algorithm can be difficult to parse on first reading, but the revision has substantially improved this point thanks to added visuals and worked-out examples. The paper has a very broad set of convincing experiments, including toy tasks, real-world datasets, and multiple baselines, as well as a discussion on the interpretability of the extracted meta-paths.